
# Climate-driven chemistry and aerosol feedbacks in CMIP6 Earth system models

Gillian Thornhill[1], William Collins[1], Dirk Olivié[2], Alex Archibald[3,4], Susanne Bauer[5], Ramiro Checa-Garcia[6], Stephanie Fiedler[7], Gerd Folberth[8], Ada Gjermundsen[2], Larry Horowitz[9], Jean-Francois Lamarque[10], Martine Michou[11], Jane Mulcahy[8], Pierre Nabat[11], Vaishali Naik[9], Fiona M. O'Connor[8], Fabien Paulot[9], Michael Schulz[2], Catherine E. Scott[12], Roland Seferian[11], Chris Smith[12], Toshihiko Takemura[13], Simone Tilmes[10], James Weber[3],

[1]Department of Meteorology, University of Reading, Reading, RG6 6BB, UK
[2]Norwegian Meteorological Institute, Oslo, Norway
[3]Department of Chemistry, University of Cambridge, Cambridge, CB2 1EW, UK
[4]National Centre for Atmospheric Science, UK
[5]NASA Goddard Institute for Space Studies, 2880 Broadway
[6] IPSL/LSCE CEA-CNRS-UVSQ-UPSaclay UMR Gif sur Yvette, FRANCE
[7]Max-Planck-Institute for Meteorology, Hamburg, 20146, Germany
[8]Met Office Hadley Centre, Exeter, EX1 3PB, United Kingdom
[9] GFDL/NOAA, Princeton University, Princeton, NJ 08540-6649
[10] National Centre for Atmospheric Research, Boulder, CO, USA
[11] Centre National de Recherches Météorologiques, Meteo-France, Toulouse Cedex, France
[12]School of Earth and Environment, University of Leeds, Leeds, LS2 9JT
[13] Research Institute for Applied Mechanics, Kyushu University, Fukuoka, Japan
*Correspondence to*: Bill Collins (w.collins@reading.ac.uk)

**Abstract.**

Feedbacks play a fundamental role in determining the magnitude of the response of the climate system to external forcing, such as from anthropogenic emissions. The latest generation of Earth system models include aerosol and chemistry components that interact with each other and with the biosphere. These interactions introduce a complex web of feedbacks which it is important to understand and quantify.

This paper addresses the multiple pathways for aerosol and chemical feedbacks in Earth system models. This is achieved by extending previous formalisms which include $CO_2$ concentrations as a state variable to a formalism which in principle includes the concentrations of all climate-active atmospheric constituents. This framework is demonstrated by applying it to the Earth system models participating in CMIP6 with a focus on the non-$CO_2$ reactive gases and aerosols (methane, ozone, sulphate aerosol, organic aerosol and dust).

We find that the overall climate feedback through chemistry and aerosols is negative in the CMIP6 Earth system models due to increased negative forcing from aerosols with warmer temperatures. Through diagnosing changes in methane emissions and lifetime we find that if Earth system models were to allow methane to vary interactively, methane positive feedbacks (principally wetland methane emissions and biogenic VOC emissions) would offset much of the aerosol feedbacks.



## 1 Introduction

Earth system models extend the complexity of climate models by representing land and ocean biospheres, atmospheric chemistry and aerosols. Within these models, natural processes, chemical reactions and biological transformations respond to changes in climate; and these processes in turn affect the climate. Therefore, the physical climate system and the biogeochemical cycles are coupled, leading to climate feedbacks that may act to further amplify or dampen the climate response to a climate forcing (Ciais et al., 2013; Heinze et al., 2019). The importance of biogeochemical feedbacks has long been

recognised for the longer timescales involved in paleoclimate studies, but the realisation of their relevance in the context of anthropogenic climate change is more recent. A multitude of biogeochemical feedbacks have been identified but the evaluation of their importance for future climate change remains very limited. A recent review of Earth system feedbacks (Heinze et al., 2019) examined the extensive range of feedbacks possible in an Earth system framework. Arneth et al. (2010) considered a range of terrestrial biogeochemical feedbacks interacting with the carbon cycle. O'Connor et al., (2010) reviewed potential

feedbacks involving methane. Carslaw et al. (2010) reviewed climate feedbacks involving natural and anthropogenic aerosols. Climate change can impact both the source strength of natural aerosols such as sea-salt, dust, biomass burning aerosols, or their precursors (dimethylsulphide (DMS), biogenic volatile organic compounds) and the lifetime of natural and anthropogenic aerosols through changes in transport and dry and wet deposition (Bellouin et al., 2011; Raes et al., 2010). Here we focus especially on those feedbacks that are mediated through changes in the abundances of reactive gases and aerosols, using data

from CMIP6 (Coupled Model Intercomparison Project 6) (Eyring et al., 2016) Earth system models that conducted the AerChemMIP (Aerosols and Chemistry Model Intercomparison Project) simulations (Collins et al., 2017).

Note that in this paper we use change in global mean surface temperature as our measure of climate change and for simplicity assume changes in other climate variables are proportional to this.

In section 2 we describe the theoretical framework used to diagnose the feedbacks. In section 3 we describe how the different

Earth System models implement the biogeochemical processes. Section 4 quantifies the feedbacks as implemented in the models. Section 5 compares these results with previous modelling and theoretical studies. Section 6 concludes. Supplementary material contains further details of the models used, and additional figures to support the process analysis of responses to dust and BVOCs.

## 2 Theoretical framework to analyse biogeochemical feedbacks

### 2.1 Theory

In order to compare climate feedbacks we need to compare them on a common scale of the change in the top of atmosphere radiation balance following a unit warming (in W m$^{-2}$ K$^{-1}$) (e.g. Gregory et al., 2009). Following Gregory et al. (2004) the radiative imbalance $\Delta N$ from an imposed forcing $\Delta F$ is given by $\Delta N = \Delta F + \alpha \Delta T$ where $\Delta T$ is the global mean change in surface temperature and $\alpha$ is the climate feedback parameter ($= \frac{d\Delta N}{d\Delta T}$). The total derivative $\frac{d\Delta N}{d\Delta T}$ can be split into a set of partial





derivatives $\frac{d\Delta N}{d\Delta T} = \sum_i \frac{\partial \Delta N}{\partial \Delta C_i} \frac{\partial \Delta C_i}{\partial \Delta T} = \sum_i \alpha_i$ , where the $\alpha_i$ are the individual feedback terms due to a change in a climate variable

$C_i$. For feedbacks involving changes in composition, the $\Delta C_i$ can represent changes in reactive gas or aerosol burdens or

emissions. $\alpha_i = \frac{\partial \Delta N}{\partial \Delta C_i} \frac{\partial \Delta C_i}{\partial \Delta T}$ can then be expressed as $\phi_i \gamma_i$, where $\phi_i$ is the radiative efficiency of the species per burden (Wm$^{-2}$

Tg$^{-1}$) or per emission (Wm$^{-2}$ (Tg yr$^{-1}$)$^{-1}$), and $\gamma_i$ is the change in species burden or emission with climate (Tg K$^{-1}$ or Tg yr$^{-1}$ K$^{-1}$).

The radiative efficiencies are based on effective radiative forcing (ERF) (Myhre et al., 2013) to include rapid adjustments to

changes in composition.

### 2.2 Applying the theory to Earth system models

With Earth system models, the $\phi_i$ and $\gamma_i$ coefficients can be diagnosed from idealised simulations in which only climate or

composition are changed. Here we use the set of simulations specified under the CMIP6 project (Eyring et al., 2016).

The $\gamma_i$ are diagnosed from a pair of idealised climate change scenarios, *piControl* where composition is maintained at a level

representative of 1850 conditions, and *abrupt-4xCO2* where CO$_2$ concentrations are abruptly quadrupled, but no other species

are directly perturbed. We take the 30-year time means from years 121-150 of these simulations for both the surface

temperature change and the burden/emission changes. The global mean surface temperature changes are therefore not the same

as the equilibrium climate sensitivities (ECSs) derived from the *abrupt-4xCO2* but are temperatures consistent with the

averaging period for the burden or emissions. The $\gamma_i$ are calculated from the change in emission or burden divided by the

temperature change.

The $\phi_i$ coefficients for changes in emissions are derived from pairs of AerChemMIP simulations, *piClim-control* where

composition and climate are maintained at a level representative of 1850 conditions, and experiments *piClim-2x* (table 1) in

which individual natural emission fluxes are doubled. The climate change in these simulations is restricted by using fixed sea

surface temperatures and sea ice cover (Collins et al., 2017) for a 30-year mean of the *piControl* simulation. The ERFs are

determined by the mean difference in top of atmosphere radiative fluxes between the *piClim-2x* and the *piClim-control* over a

30-year period. The $\phi_i$ are calculated from the ERF divided by either the change in emission or change in burden, depending

on the units of $\gamma_i$ above.



| Experiment | Flux to be doubled |
|---|---|
| *piClim-control* | None |
| *piClim-2xdust* | Dust |
| *piClim-2xss* | Sea salt |
| *piClim-2xDMS* | Oceanic DMS |
| *piClim-2xNOX* | Lightning $NO_X$ |
| *piClim-2xVOC* | Biogenic VOCs |

**Table 1: List of simulations for diagnosing ERFs of natural emitted species. The specified natural emission fluxes are doubled**
**compared to the 1850 control.**

For $\phi_{O_3}$ a radiative efficiency of 0.042 W m$^{-2}$ per Dobson Unit (DU) is used in the troposphere (Stevenson et al., 2013).

The ESM setups here, even with tropospheric chemistry, still constrain methane to specified concentrations at the surface. This means that any feedbacks mediated through changes in oxidising capacity have a negligible effect on methane. It is however possible to diagnose the change in methane that would be expected, if it were not constrained, from the change in its lifetime

$\frac{\Delta C}{C} = \left(\frac{\Delta \tau}{\tau} + 1\right)^f - 1 \approx f\frac{\Delta \tau}{\tau}$, where C is the methane concentration, $\tau$ is the total methane lifetime (including loss to soils) and $f$ is the feedback of methane on its own lifetime (Fiore et al., 2009). The radiative forcing from the change in concentration is calculated using the formula from Etminan et al., (2016) scaled by 1.65 to account for change in ozone and stratospheric water vapour (Myhre et al., 2013). This gives 1.15 W m$^{-2}$ per fractional change in methane lifetime (based on 1850 baseline concentrations of methane and N$_2$O). Changes in methane concentration due to changes in emissions $\Delta E$ are given by $\Delta C =$

$\Delta E \tau f \left(\frac{m_{\text{air}}}{m_{\text{CH}_4}}\right) / M_{\text{atm}}$ , where $\tau$=9.25 years, and  $f$=1.34 (Myhre et al., 2013). $m_{\text{air}}$ and $m_{\text{CH}_4}$ are the relative molecular masses of air and methane.

## 3 Model descriptions

### 3.1 Model implementation of aerosols, tropospheric and stratospheric chemistry

We use results from 6 Earth system models that contributed simulations under the AerChemMIP *piClim-2x* experimental setup.
All six models have interactive aerosol schemes, four have interactive stratospheric chemistry and three have interactive tropospheric chemistry. The level of sophistication of the chemistry can affect the modelled responses to the emissions of reactive gases. For instance, in models without interactive tropospheric chemistry changes in biogenic volatile organic compound emissions (BVOCs) affect only organic aerosols, whereas in models with interactive tropospheric chemistry they also affect ozone, methane lifetime, and potentially the oxidation of other aerosol precursors.






| | Tropospheric chemistry | Stratospheric chemistry | Reference |
|---|---|---|---|
| NorESM2 | No | No | (Olivié et al., in prep; Kirkevåg et al., 2018)) |
| UKESM1 | Interactive | Interactive | (Archibald et al., 2019; Sellar et al., 2019) |
| CNRM-ESM2 | No | Interactive | (Séférian et al., 2019) |
| MIROC6 | No | No | (Tatebe et al., 2019) |
| GFDL-ESM4 | Interactive | Interactive | (Horowitz et al., in prep) |
| CESM2-WACCM | Interactive | Interactive | (Gettelman et al., 2019) |

**Table 2 Sophistication of gas-phase chemistry used in the Earth system models (For further details see** (Thornhill et al., submitted) **.**

### 3.2 Model implementation of natural emissions

#### 3.2.1 Land

The principle land-based natural emissions are dust and BVOCs (table 3).

Dust emissions are parameterised as a function of surface wind speeds or wind stress, and account for the amount of bare soil, soil type, and aridity (Ackerley et al., 2012; Collins et al., 2011; Evan et al., 2014; Fiedler et al., 2016; Huneeus et al., 2011; Shao et al., 2011; Zender et al., 2004). There is a variation between the models in the sizes considered, whether binned or modal, and the optical properties of the dust particles (Kok et al., 2018; Xie et al., 2018). Table S1 lists the parameterizations for desert-dust aerosol for the contributing models and the simulated dust-aerosol sizes.

BVOC emissions are parametrised as a function of vegetation type and cover, and also temperature and photosynthesis rates (gross primary productivity) (Guenther, 1995; Pacifico et al., 2011; Sporre et al., 2019; Unger, 2014). Some parameterisations also include dependence on $CO_2$ concentrations (Pacifico et al., 2012). Models differ in the speciation of the VOCs emitted but typically include isoprene and monoterpenes, with different emission parameterisations for different species. The ability of VOCs to form secondary organic aerosol are typically parameterised as a fixed yield (Mulcahy et al., 2019). For further

details see table S1 and references therein.





|  | Dust | VOC |
|---|---|---|
| NorESM2 | LAI varies | Dependence on PAR, temperature |
| UKESM1 | Interactive vegetation | Dependence on PAR, temperature, vegetation |
| CNRM-ESM2 | Prescribed land cover | Prescribed |
| MIROC6 | LAI varies | Prescribed |
| GFDL-ESM4 | Interactive vegetation | Dependence on PAR, temperature. Not dependent on vegetation. |
| CESM2-WACCM | LAI varies | Dependence on PAR, temperature |

**Table 3 Levels of complexity of vegetation included in the land-based emssions schemes of dust and BVOCs for the ESMs.**

### 3.2.2 Marine

The principle ocean emissions are of sea salt, dimethyl-sulphide (DMS) and primary organic aerosols (table 4).

The air-sea exchange processes for these emissions are parameterised as a function of wind speed and sometimes temperature (Gong, 2003; Jaeglé et al., 2011).

Changes in DMS emissions can be initiated by various factors such as changes in temperature, insolation, depth of the ocean-mixed layer, sea-ice extent, wind strength, nutrient recycling, or shift in marine ecosystems (Heinze et al., 2019).  The surface sea water concentrations of DMS are prescribed in some models (CNRM-ESM2, GFDL-ESM4, MIROC6, CESM2-WACCM)

and calculated interactively from ocean biogeochemistry in others (UKESM1, NorESM2). Oceanic organic aerosol emissions are also wind-speed dependent and in addition depend on chlorophyll concentrations generated either from interactive biogeochemistry or observation-based chlorophyll concentrations in models without ocean biogeochemistry components.





| | Sea salt | DMS | Oceanic organic aerosol |
|---|---|---|---|
| NorESM2-LM | Temperature and wind dependent | Interactive biogeochemistry for sea water DMS concentration, wind speed and temperature for air-sea DMS flux | Interactive biogeochemistry for Chlorophyll concentrations; wind speed; sea salt emission flux |
| UKESM1 | Wind speed | Interactive biogeochemistry | Interactive biogeochemistry |
| CNRM-ESM2-1 | Temperature, wind speed | Prescribed emissions | |
| MIROC6 | Wind speed | Dependent on wind speed and chlorophyll | Dependent on wind speed and chlorophyll |
| GFDL-ESM4 | Temperature, wind speed | Wind speed. Prescribed sea water concentration | |
| CESM2-WACCM | Wind speed | Wind speed. Prescribed sea water concentration | Wind speed |

**Table 4 Levels of complexity of marine emissions in the ESMs**

### 145 3.2.3 Lightning

The models with tropospheric chemistry (UKESM1, GFDL-ESM4, CESM2-WACCM) all include parameterisations of the emission of nitrogen oxides ($NO_X$) from lightning, related to the height of the convective cloud top (Price et al., 1997; Price and Rind, 1992). The lightning frequency depends strongly on the convective cloud top height, and the ratio of cloud-to-cloud versus cloud-to-ground lightning depends on the cold cloud thickness (from 0°C to the cloud top).

### 150 4 Quantification of feedbacks

The feedbacks in this section are all derived from the difference between the *piControl* and *abrupt-4xCO2* CMIP6 experiments. The Earth system models all respond with different levels of climate change, so all climate feedbacks are normalised to the change in global mean surface temperature between *abrupt-4xCO2* and *piControl* for the 30-year period years 121-150 (table 5). There is a factor of nearly two between the temperature responses of the models. Since this timeframe is not long enough for the models to have reached equilibrium these temperatures are not the same as equilibrium climate senstivity (ECS).





|  | CNRM-ESM2 | UKESM1 | MIROC6 | NorESM2 | CESM2-WACCM | GFDL-ESM4 |
|---|---|---|---|---|---|---|
| ΔT 4xCO2 (K) | 6.02 | 7.46 | 4.01 | 3.96 | 6.37 | 3.93 |

**Table 5: Change in global mean surface temperature following an abrupt quadrupling of CO₂ concentrations. Difference between** *abrupt-4xCO2* **and** *piControl* **averaged over the years 121-150.**

## 4.1 Aerosol species

### 4.1.1 Desert Dust

The *2xdust* perturbation is applied by scaling the parameterisation in the emission scheme. Since changing dust emissions will affect the boundary layer meteorology the net effect is not an exact doubling of the emissions (table 6). Four of the five models in AerChemMIP have a negative radiative forcing for doubled dust aerosols as the negative shortwave radiative effects outweigh the positive longwave radiative effects of dust aerosols (figures 1(a), S1(a-e); table 6). The only exception is CNRM-ESM2, where the global shortwave forcing is also positive, explaining the different sign of the ERF compared to the other models. The ERF for GFDL-ESM4 is not significantly different from zero. UKESM1 has by far the largest dust emissions (and change from doubling) because it includes particles that are emitted and deposited in the same timestep. CNRM-ESM2 also includes large particles (up to 50μm). These models however have similar changes in dust aerosol optical depth (AOD) compared to the other models and hence the magnitude of the forcing efficiency per change in AOD is not out of line with the others. MIROC6 has the strongest forcing even with the lowest emissions and smallest change in AOD, thus giving it the largest forcing efficiency per AOD (figure S1 (f-j)).

The response of dust aerosols to *abrupt-4xCO2* (figure 1(b)) is substantially different across the model ensemble. Three models (CNRM-ESM2, MIROC6 and GFDL-ESM4) show an increase in dust emission in a *4xCO2* climate due to increased aridity and near-surface wind speeds, whereas UKESM has a decrease in dust emissions with more $CO_2$ due to increased fertilisation of the vegetation (hence less bare soil) paired with decreased near-surface winds. NorESM2 shows near zero change. The spatial pattern of the opposing response of dust emission to *4xCO2* in the two most extreme models, UKESM1 and CNRM-ESM2, is consistent with the responses in 10m-wind speed to *4xCO2* (figure S2). These clearly reflect larger (smaller) increases in mean winds over regions where the mean emission amount is larger (smaller) for *4xCO2* compared to the pre-industrial climatology. The increase or decrease in winds is also likely to be affected by changes in vegetation in semi-arid regions, e.g., the Sahel.

As well as affecting the emissions, changing climate can also affect the removal of dust. In all models except UKESM1 the lifetime of dust increases. The effect of an increase in lifetime can be seen by comparing the change in AOD. The modelled changes in dust AOD in the *abrupt-4xCO2* experiment are one and a half to twice as large (for those models where lifetime increases) as would be expected assuming a linear scaling with emissions across all size ranges ("scaled AOD" in table 6).





The climate feedback parameter for dust is given by the product of the radiative efficiencies ($\phi$) with the sensitivities to climate ($\gamma$). These vary from -0.016 to +0.048 W m$^{-2}$ K$^{-1}$ with a multi-model mean of -0.003±0.008 W m$^{-2}$ K$^{-1,}$ i.e. averaging to a value near zero. Although some models obtain similar feedback terms, this is not necessarily for the same reason. For instance, CNRM-ESM2 and UKESM1 have a positive dust feedback, though for opposite reasons; an increase in positive forcing in CNRM-ESM2 and a decrease in negative forcing in UKESM1. For instance, NorESM2 has a large ERF for doubled dust

emissions, the small change in dust emission for *4xCO2*, however, does not lead to a large feedback for that model.

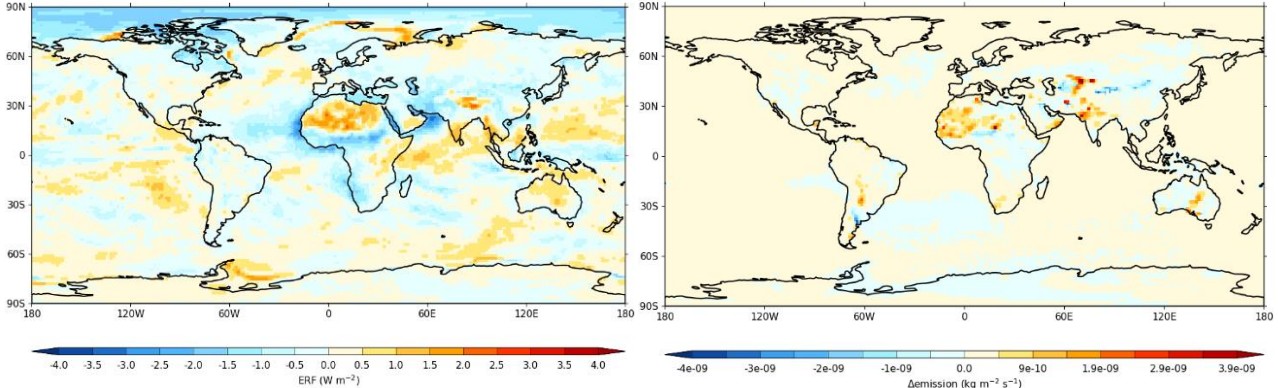

**Figure 1 Multi-model mean (a) ERF from *piClim-2xdust* vs *piClim-control*, (b) Change in dust emissions for *abrupt-4xCO2* vs *piControl*.**





| | CNRM-ESM2 | UKESM1 | MIROC6 | NorESM2 | GFDL-ESM4 | Multi-model |
|---|---|---|---|---|---|---|
| Emission *control* Tg yr⁻¹ | 2750 | 8066 | 1106 | 1661 | 1981 | |
| ΔEmission *2xdust* Tg yr⁻¹ | 2877 | 8256 | 1065 | 1397 | 1989 | |
| ERF *2xdust* W m⁻² | 0.09±0.03 | -0.07±0.03 | -0.18±0.04 | -0.14±0.07 | -0.00±0.03 | |
| ERF/ Emission W m⁻² (Tg yr⁻¹)⁻¹ | 3.1±1.0E-5 | -8.9±3.6E-6 | -1.7±0.4E-4 | -1.1±0.5E-4 | -0.2±1.5E-5 | |
| ERF/AOD W m⁻² | 8.0±-2.7 | -5.8±2.4 | -25.6±5.6 | -6.0±2.8 | -0.2±1.6 | |
| ΔEmission/ΔT Tg yr⁻¹ K⁻¹ | 65±4 | -109±15 | 70.±7 | -9±0 | 181±10 | |
| Δlifetime/ΔT % K⁻¹ | 2.6±0.2 | -0.4±0.4 | 1.9±0.9 | 1.2±0.3 | 3.7±0.6 | |
| scaled AOD/ΔT K⁻¹ | 2.5±0.2E-4 | -1.7±0.2E-4 | 4.8±0.4E-4 | -1.6±E-4 | 17.3±1.0E-4 | |
| *4xCO2* ΔAOD/ΔT K⁻¹ | 6.0±0.3E-4 | -2.6±0.6E-4 | 6.3±0.5E-4 | | 26.5±1.3E-4 | |
| α emissions W m⁻² K⁻¹ | 0.0020±0.0007 | 0.0010±0.0004 | -0.012±0.003 | 0.0010±0.0005 | -0.0004±0.0027 | -0.002±0.005 |
| α AOD W m⁻² K⁻¹ | 0.0048±0.0016 | 0.0015±0.0007 | -0.016±0.004 | | -0.0006±0.0042 | -0.003±0.008 |

**Table 6. Dust radiative efficiencies by emission and AOD from *2xdust* experiments. Changes in emission and AOD from *abrupt-4xCO2*. "scaled" refers to scaling the *2xdust* relations between AOD and emissions by the *4xCO2* changes in emissions. alpha values are calculated assuming ERF is proportional to emissions or AOD. Uncertainties for each model are errors in the mean based on interannual variability. Uncertainties in the multi-model results are standard deviation across the models.**

#### 4.1.2 Sea Salt

All models show a strong negative forcing to double sea salt emissions (figure 2(a), table 7), although the ERF for MIROC6 is significantly smaller than the others. The emissions and mass loading for the CNRM-ESM2 model are approximately twenty times those of the other models, although the AOD change is similar to other models. All models show a similar forcing efficiency per AOD change. All models show an increase in sea salt emissions in the Southern Ocean in *4xCO2* (figure 2(b)) due to increased wind speeds, with a general tendency for decreases elsewhere due to rising temperatures (Jaeglé et al., 2011).

The global mean change in emissions is positive in all models except MIROC6. For models showing an increased lifetime in 4xCO2 the modelled increase in AOD is larger than that expected from scaling the emissions change. Although emissions (and




the mass burden) of sea salt decrease in MIROC6 the AOD increases. The mean feedback is -0.031±0.31 W m$^{-2}$ K$^{-1}$ based on emissions, and -0.060±0.56 W m$^{-2}$ K$^{-1}$ based on the increase in AOD. The signs are consistently negative except for the emission-based feedback for MIROC6.

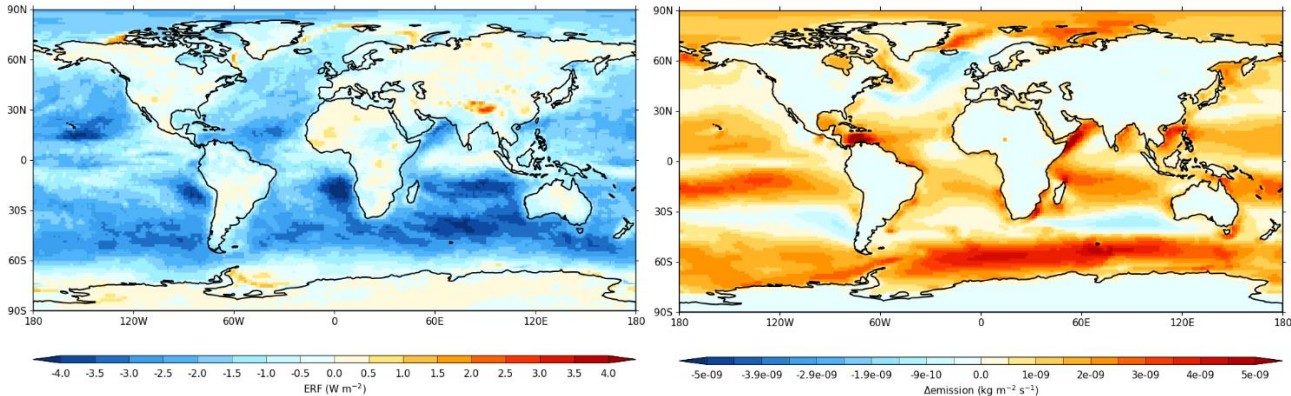


**Figure 2 Multi-model mean (a) ERF from *piClim-2xss* vs *piClim-control*, (b) Change in sea-salt emissions for *abrupt-4xCO2* vs *piControl*.**



| | CNRM-ESM2 | UKESM1 | MIROC6 | NorESM2 | GFDL-ESM4 | Multi-model |
|---|---|---|---|---|---|---|
| ΔEmission *2xss* Tg yr$^{-1}$ | 64939 | 5500 | 3577 | 3771 | 5675 | |
| ERF *2xss* W m$^{-2}$ | -1.04±0.03 | -1.29±0.03 | -0.35±0.04 | -2.28±0.07 | -1.84±0.03 | |
| ERF/ Emission W m$^{-2}$ (Tg yr$^{-1}$)$^{-1}$ | -1.61±0.04 E-5 | -2.30±0.05E-4 | -9.72±1.12E-5 | -6.0±0.2E-4 | -3.20±0.07 E-4 | |
| ERF/AOD W m$^{-2}$ | -19.8±0.6 | | -25±3 | -26±0.8 | -38.7±0.8 | |
| ΔEmission/ΔT Tg yr$^{-1}$ K$^{-1}$ | 2570±87 | 6.0±2.6 | -3.93±2.6 | 72±4 | 258±9 | |
| Δlifetime/ΔT % K$^{-1}$ | 0.44±0.13 | -0.20±0.06 | -0.68±0.09 | -0.92±0.14 | 1.8±0.2 | |
| Scaled AOD/ΔT K$^{-1}$ | 20.8±0.7E-4 | | -0.16±0.10E-4 | 17±1E-4 | 21.6±0.8E-4 | |
| *4xCO2* ΔAOD/ΔT K$^{-1}$ | 24.8±0.8E-4 | | 0.62±0.20E-4 | | 33.6±1.0E-4 | |
| α emissions W m$^{-2}$ K$^{-1}$ | -0.041±0.002 | -0.0014±0.0006 | 0.0004±0.0003 | -0.044±0.003 | -0.084±0.004 | -0.031±0.031 |
| α AOD W m$^{-2}$ K$^{-1}$ | -0.049±0.002 | | -0.0015±0.0005 | | -0.130±0.005 | -0.060±0.053 |

**Table 7. Radiative efficiencies by emission and AOD from *2xss* (sea-salt). Changes in emission and AOD from *4xCO2*. "scaled" refers to scaling the *2xss* relations between AOD to emissions by the *4xCO2* changes in emissions. α values are calculated assuming ERF is proportional to emissions or AOD. Uncertainties for each model are errors in the mean based on interannual variability. Uncertainties in the multi-model results are standard deviation across the models. Not all models provided AOD diagnostics.**

### 4.1.3 DMS

All models except CNRM-ESM2 have interactive DMS emissions that vary with climate (wind speed), and two also include interactive ocean biogeochemistry (UKESM1 and NorESM2). The latter two performed the *2xDMS* experiment. CNRM-ESM2 also ran the *2xDMS* experiment but uses prescribed emissions that are independent of climate. The ERF for *2xDMS* is negative for all three models that ran this experiment (figure 3(a)), though less strongly so for CNRM-ESM2. Three of the models with interactive emissions show a decrease in sulphur emissions in *4xCO2* where the tropical decrease more than compensates for the increase along the edge of the sea ice retreat. GFDL-ESM4 shows an increase in overall sulphur emissions. The multi-model mean is shown in figure 3(b). Since not all data is available for all models, we use the multi-model radiative efficiencies (by emission and by mass) and the multi-model sensitivities (of emissions and mass) to climate in order to calculate the multi-model feedback (table 8). The strong positive DMS increase in GFDL-ESM4 weakens the multi-model mean decrease in emission with climate. The multi-model mean emission-based α is therefore near-zero (within the uncertainty




range). There is an increased sulphur mass in all models in the *4xCO2* simulation due to an increase in the sulphate lifetime.
When scaled by the radiative efficiency for DMS emissions (which might not be appropriate for a lifetime increase) this leads
to negative α (-0.048±0.028 W m$^{-2}$ K$^{-1}$).

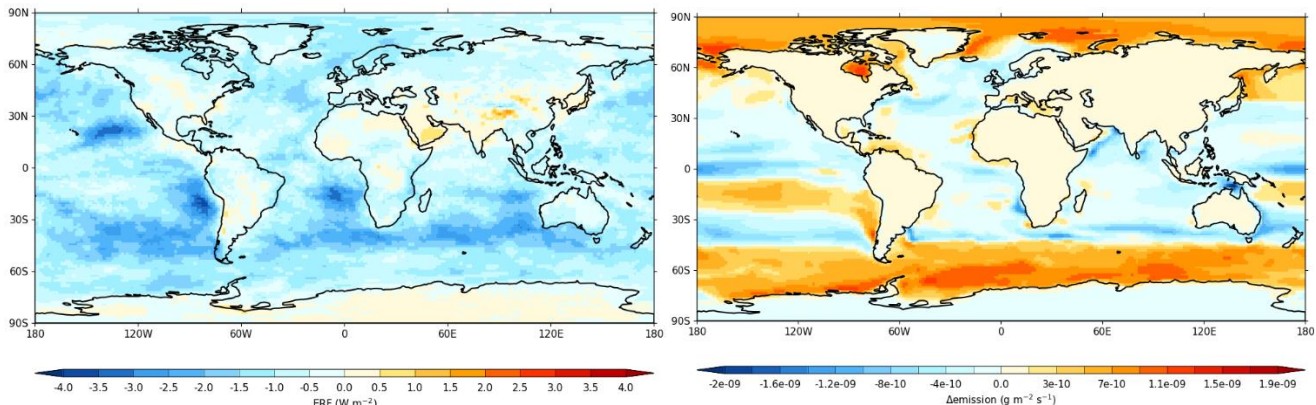

**Figure 3 Multi-model mean (a) ERF from *piClim-2xDMS* vs *piClim-control*, (b) Change in DMS emissions for *abrupt-4xCO2* vs *piControl***





| | CNRM-ESM2 | UKESM1 | MIROC6 | NorESM2 | GFDL-ESM4 | Multi-model |
|---|---|---|---|---|---|---|
| ERF *2xDMS* W m$^{-2}$ | -0.37±0.03 | -1.22±0.03 | | -1.27±0.07 | | |
| ERF/ Emission W m$^{-2}$ (Tg(S) yr$^{-1}$)$^{-1}$ | -0.0089±0.0007 | -0.0386±0.0010 | | -0.0348±0.0019 | | -0.027±0.013 |
| ERF/mass W m$^{-2}$ | -3.8±0.3 | -3.15±0.08 | | -5.4±0.3 | | -4.1±1.0 |
| ΔEmission/ΔT Tg(S) yr$^{-1}$ K$^{-1}$ | | -0.07±0.02 | -0.014±0.004 | -0.36±0.03 | 0.39±0.03 | -0.01±0.24 |
| Scaled mass/ΔT Tg(S) K$^{-1}$ | | -8.9±2.1E-4 | | -23±2E-4 | | |
| *4xCO2* Δmass/ΔT Tg(S) K$^{-1}$ | 25.4±0.9E-4 | 186±7E-4 | 59±2E-4 | 14±6E-4 | 172±7E-4 | 120±6E-4 |
| Δlifetime/ΔT % K$^{-1}$ | 1.98±0.4 | 2.48±0.06 | 1.91±0.07 | 2.73±0.11 | 2.42±0.09 | |
| α emissions W m$^{-2}$ K$^{-1}$ | | 0.0028±0.0007 | | 0.0125±0.0013 | | 0.000±0.007 |
| α mass W m$^{-2}$ K$^{-1}$ | -0.0097±0.0009 | -0.059±0.003 | | -0.075±0.005 | | -0.048±0.028 |

**Table 8. Radiative efficiencies by emission and mass from *2xDMS*. Changes in emission and mass from *4xCO2* experiment. Emissions are for DMS or SO$_2$+SO$_4$ depending on the model. "scaled" refers to scaling the *2xDMS* relations between mass and emissions by the *4xCO*2 changes in emissions. α values are calculated assuming ERF is proportional to emissions or mass. Multi-model mean values of α use the multi-model mean radiative efficiencies and sensitivities to climate, rather than being an average of the individual model α values. Uncertainties for each model are errors in the mean based on interannual variability. Uncertainties in the multi-**
**model results are standard deviation across the models. MIROC6 and GFDL-ESM4 did not perform the *2xDMS* experiment, but DMS changes are diagnosed from their *4xCO2* experiments. DMS emissions do not vary in the CNRM-ESM2 *4xCO2* experiment.**

### 4.1.4 Organic aerosol

Biogenic VOC emissions lead to both organic aerosol and ozone production (in those models with tropospheric chemistry). It is therefore difficult to distinguish the two in the ERFs in these models. An estimate of the direct aerosol effect can be
determined by additional radiation diagnostics that are run without the contribution of aerosols "aerosol-free" (ERF$_{af}$), for clear sky conditions (ERF$_{cs}$), and both clear sky and aerosol free (ERF$_{csaf}$) (Ghan, 2013). Here the aerosol direct effect is ERF-ERF$_{af}$ and the cloud effect is ERF$_{af}$-ERF$_{csaf}$ (although this may include cloud forcing due to adjustments caused by the ozone changes too). The direct aerosol and cloud radiative effects are shown in figure 4. These estimated aerosol forcing changes are significant (between -0.3 and -0.69 W m$^{-2}$). The most negative forcing comes from the NorESM2 model which has no changes
in gas-phase chemistry (table 9).





In terms of aerosol, there is an increase in organic aerosol (OA) mass and expected increase in AOD with very similar spatial pattern when the emission of BVOCs is doubled. Changes to cloud droplet number concentration are more complex and may not be spatially co-located with the changes to BVOC emission (figure S4). Whilst the additional secondary organic aerosol can grow particles to a size where they can act as cloud condensation nuclei, this process can also enhance the aging rate of

particles removing them from the atmosphere more quickly. In addition, for models with interactive tropospheric chemistry, the decrease in oxidant concentrations resulting from a doubling of VOC emissions can prevent the oxidation of sulphur containing species that might otherwise have formed aerosols, leading to a reduction in CDNC.  In the *4xCO2* experiments, these models also simulate an increase in primary organic aerosol emission from the ocean which adds to the OA mass on top of the effect of BVOC emissions. The feedback factors are negative and very strong in some models, ranging from -0.003 to -

0.276 W m$^{-2}$ K$^{-1}$ based on emissions and -0.025 to -0.359 W m$^{-2}$ K$^{-1}$ based on mass assuming all OA has the same radiative efficiency as that from vegetation.

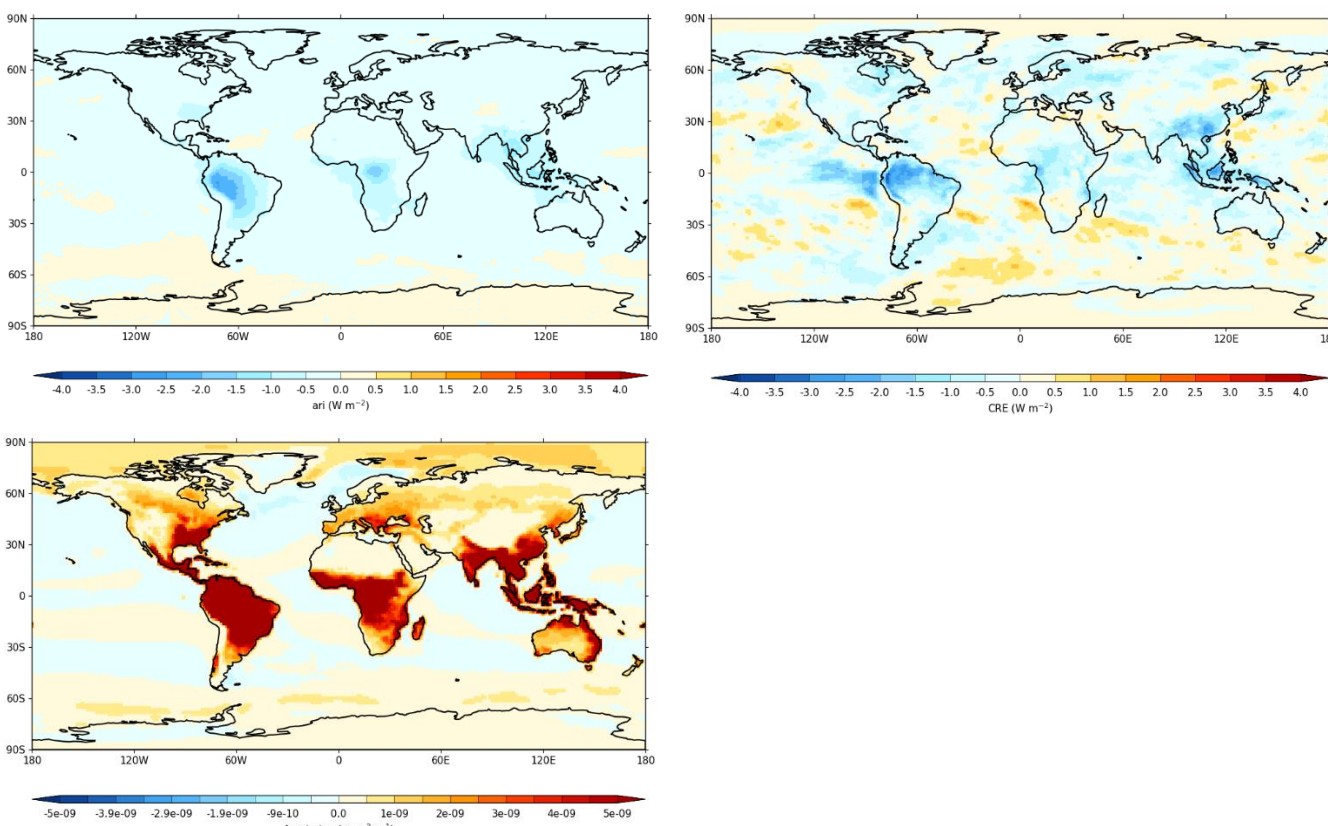

**Figure 4 Multi-model mean (a) Aerosol direct effect from *piClim-2xVOC* vs *piClim-control*, (b) cloud radiative effect from *piClim-**

***2xVOC* vs *piClim-control*, (c) Change in organic aerosol emissions for *abrupt-4xCO2* vs *piControl*.**



| | UKESM1 | | NorESM2 | | GFDL-ESM4 | | CESM2-WACCM | Multi-model |
|---|---|---|---|---|---|---|---|---|
| ERF *2xVOC* W m$^{-2}$ | direct | cloud | direct | cloud | direct | cloud | -0.31 | |
| | -0.19± 0.03 | 0.12± 0.03 | -0.18± 0.07 | -0.55± 0.07 | -0.21± 0.04 | -0.30± 0.04 | | |
| ERF/ Emission W m$^{-2}$ (Tg) yr$^{-1}$)$^{-1}$ | -1.0±0.4E-4 | | -11.8±1.2E-4 | | -10.9±0.6E-4 | | -4.7±0.6E-4 | -7±4E-4 |
| ERF/mass W m$^{-2}$ | -0.19±0.08 | | -0.56±0.06 | | -0.70±0.04 | | -0.29±0.04 | -0.34±0.16 |
| ΔEmissionVOC/ΔT Tg yr$^{-1}$ K$^{-1}$ | 33 | | 234±7 | | 81±2 | | 150±2 | 124±76 |
| Scaled mass/ΔT Tg K$^{-1}$ | 0.017 | | 0.50±0.02 | | 0.127±0.0.003 | | 0.243±0.003 | |
| *4xCO2* Δmass/ΔT Tg K$^{-1}$ | 0.135±0.004 | | 0.644±0.018 | | 0.022±0.001 | | 0.510±0.018 | 0.33±0.26 |
| Δlifetime/ΔT % K$^{-1}$ | 1.1±0.2 | | 33±6 | | -4.2±0.2 | | | |
| α emissions W m$^{-2}$ K$^{-1}$ | -0.003 | | -0.28±0.03 | | -0.089±0.006 | | -0.070±0.009 | -0.088±0.077 |
| α mass W m$^{-2}$ K$^{-1}$ | -0.025±0.01 | | -0.359±0.03 | | -0.015±0.001 | | -0.148±0.02 | -0.113±0.102 |

**Table 9.** Radiative efficiencies by emission and mass from *2xVOC*. Changes in emission and mass from *4xCO2* experiment. "scaled" refers to scaling the *2xVOC* relations between mass and emissions by the 4xCO2 changes in emissions. α values are calculated assuming ERF is proportional to emissions or mass. Multi-model mean values of α use the multi-model mean radiative efficiencies and sensitivities to climate, so are different to the average of the individual model α values. Uncertainties for each model are errors in the mean based on interannual variability. Uncertainties in the multi-model results are standard deviation across the models.

## 4.2 Gas-phase feedbacks

### 4.2.1 Biogenic VOCs

To estimate the stratospheric-temperature adjusted radiative forcing (SARF) from the ozone changes, and to remove the effect of aerosols we use the clear-sky aerosol-free ERF (ERF$_{csaf}$) (table 10). However, this neglects any cloud adjustments caused by the ozone, and any cloud masking of the direct ozone SARF. For GFDL-ESM4 and CESM2-WACCM the magnitude of the ozone forcing is smaller than that for aerosols leading to a net negative ERF from BVOCs. For UKESM1 the net ERF is positive due to a lower magnitude of aerosol forcing. The ozone contribution is also estimated assuming a radiative efficiency of 0.042 W/m$^2$ per Dobson Unit (Stevenson et al., 2013). This efficiency is strictly only applicable to





changes in tropospheric ozone but is also applied to the stratospheric ozone since these changes occur in the lower stratosphere just above the tropopause. The estimated ozone SARF (tropospheric + stratospheric) is within the range of the diagnosed $ERF_{csaf}$. CESM2-WACCM has the largest BVOC emissions and a decrease in tropospheric ozone column, although a strong increase in the stratospheric column. This is likely to be due to NOx-limited chemistry near the surface and increased transport of reactive nitrogen ($NO_Y$) away from the surface to the upper troposphere and lower stratosphere as peroxy-acetyl nitrate (PAN) and other organic nitrates. The overall feedback ranges from 0.004 W m$^{-2}$ K$^{-1}$ for UKESM1 which has the lowest ozone response to BVOC emissions, and the lowest BVOC increase with climate (due to $CO_2$ inhibition) to 0.028 W m$^{-2}$ K$^{-1}$ for CESM2-WACCM which has the strongest VOC response to climate.

| | UKESM1 | | GFDL-ESM4 | | CESM2-WACCM | | Multi-model | |
|---|---|---|---|---|---|---|---|---|
| $ERF_{csaf}$ *2xVOC* W m$^{-2}$ | 0.09±0.03 | | 0.13±0.04 | | | | | |
| $ERF_{csaf}$/emission W m$^2$(Tg yr$^{-1}$)$^{-1}$ | 1.2±0.4E-4 | | 2.8±0.8E-4 | | | | | |
| | trop | strat | trop | strat | trop | strat | | |
| Ozone/emission DU (Tg yr$^{-1}$)$^{-1}$ | 0.0015 | 0.0021 | 0.0022 | 0.0031 | -0.0003 | 0.0044 | | |
| Ozone SARF /emission W m$^2$(Tg yr$^{-1}$)$^{-1}$ | 0.63±0.09 E-4 | 0.9±0.1 E-4 | 0.9±0.1 E-4 | 1.3±0.2 E-4 | -0.10±0.01 E-4 | 1.8±0.3 E-4 | | |
| *4xCO2* Tg yr$^{-1}$ K$^{-1}$ | 32.5±0.8 | | 81±2 | | 150±2 | | | |
| α $ERF_{csaf}$ W m$^{-2}$ K$^{-1}$ | 0.004±0.001 | | 0.023±0.007 | | | | | |
| α $SARF_{O3}$ W m$^{-2}$ K$^{-1}$ | 0.0021± 0.0003 | 0.0028± 0.0004 | 0.0076± 0.0011 | 0.011± 0.001 | -0.0015± 0.0002 | 0.028± 0.004 | 0.003± 0.0040 | 0.014± 0.010 |

**Table 10. Radiative efficiencies (clear-sky aerosol-free $ERF_{csaf}$) for *2xVOC* emissions. Tropospheric and stratospheric ozone column changes and their estimated radiative effects. Changes in emission from *4xCO2* experiment. α values are calculated assuming $ERF_{csaf}$ or a SARF efficiency for ozone of 0.042 W m$^{-2}$ DU$^{-1}$. Uncertainties for each model are errors in the mean based on interannual variability, and assuming a 14% uncertainty in the ozone radiative efficiency. Uncertainties in the multi-model results are standard deviation across the models.**





BVOC emissions also affect the methane lifetime. Methane does not change in the AerChemMIP experimental setup, but the methane changes that would be expected if methane were allowed to evolve freely can be diagnosed from the change in methane lifetime. We diagnose this from changes in methane loss rate throughout the atmosphere (which includes stratospheric

loss processes) and assume a loss to soils with a lifetime of 120 years (Ciais et al., 2013). All three models show similar sensitivities 0.030 to 0.047 % increase in methane lifetime per Tg $yr^{-1}$ of BVOC change due to decreases in OH in the mid-upper tropical free troposphere (not shown). From this the expected lifetime changes can be deduced from the change in BVOC emissions with temperature (table 11). These lifetime changes are converted to feedbacks using the radiative efficiency (including impacts on ozone and stratospheric water vapour) for methane lifetime changes in section 2.2 (0.015 W $m^{-2}$ $\%^{-1}$).

The feedbacks range from 0.012 to 0.081 W $m^{-2}$ $K^{-1}$ where the variability is mostly due to the different sensitivities of BVOCs to climate in the models. These are significantly stronger feedbacks than those due to ozone.

|  | UKESM1 | GFDL-ESM4 | CESM2-WACCM | Multi-model |
|---|---|---|---|---|
| $\tau_{CH_4}$/emission % $(Tg\ yr^{-1})^{-1}$ | 0.033 | 0.030 | 0.047 | |
| $\tau_{CH_4}$/$\Delta T$ % $K^{-1}$ | 1.1 | 2.4 | 7.1 | |
| $\alpha\ \tau_{CH_4}$ W $m^{-2}$ $K^{-1}$ | 0.012±0.002 | 0.028±0.004 | 0.081±0.011 | 0.041±0.030 |

**Table 11. Percentage change in methane lifetime for *2xVOC* emissions. Estimated change in lifetime following changes in BVOC emission from *4xCO2* experiment. α values are calculated assuming a radiative efficiency of 0.015 W $m^{-2}$ $\%^{-1}$. Uncertainties for each model assume a 14% uncertainty in the methane radiative efficiency. Uncertainties in the multi-model results are standard deviation**
**across the models.**

### 4.2.2 Lightning NO$_X$

Lightning NO$_X$ leads to ozone production, and changes in methane lifetime. In UKESM1 NO$_X$ is known to increase the formation of new sulphate particles (O'Connor and et al., submitted) offsetting positive ozone forcing. To separate the ozone effect, we use $ERF_{csaf}$ for UKESM1 as in section 4.2.1. The assumption of radiative efficiency of 0.042 W $m^{-2}$ $DU^{-1}$ seems to

agree with the ERF for GFDL-ESM and CESM2-WACCM (table 12). For UKESM1 $ERF_{csaf}$ is lower than expected from the ozone columns, suggesting that the clear-sky aerosol free component misses some of the ERF due to ozone.

Lightning NO$_X$ increases in UKESM1 and CESM2-WACCM but decreases slightly in GFDL-ESM4 although they all use variations on the cloud-top height schemes (Price et al., 1997; Price and Rind, 1992). Hence the feedback is positive for UKESM1 and CESM2-WACCM (0.009 and 0.011 W $m^{-2}$ $K^{-1}$), based on the ozone changes, but slightly negative for GFDL-

ESM4 (-0.001 W $m^{-2}$ $K^{-1}$).



| | UKESM1 | | GFDL-ESM4 | | CESM2-WACCM | | Multi-model | |
|---|---|---|---|---|---|---|---|---|
| ERF *2xNOX* W m$^{-2}$ | 0.12±0.03 | | 0.11±0.04 | | 0.15±0.04 | | | |
| ERF/emission W m$^2$(Tg yr$^{-1}$)$^{-1}$ | 0.018±0.004 | | 0.036±0.013 | | 0.050±0.013 | | | |
| | trop | strat | trop | strat | Trop | strat | | |
| Ozone/emission DU (Tg yr$^{-1}$)$^{-1}$ | 0.59 | 0.13 | 0.72 | 0.22 | 0.90 | 0.37 | | |
| Ozone SARF /emission W m$^2$(Tg yr$^{-1}$)$^{-1}$ | 0.025± 0.003 | 0.0055± 0.0007 | 0.030± 0.004 | 0.009± 0.001 | 0.038± 0.005 | 0.015± 0.002 | | |
| *4xCO2* Tg yr$^{-1}$ K$^{-1}$ | 0.27±0.01 | | -0.029±0.008 | | 0.208±0.006 | | | |
| α ERF$_{csaf}$ W m$^{-2}$ K$^{-1}$ | 0.005±0.001 | | -0.0010±0.0005 | | 0.010±0.003 | | | |
| α SARF$_{O3}$ W m$^{-2}$ K$^{-1}$ | 0.007± 0.001 | 0.0015± 0.0002 | -0.0009± 0.0003 | 0.000±0.001 | 0.008± 0.003 | 0.0032± 0.0005 | 0.005± 0.004 | 0.001± 0.001 |

**Table 12. Radiative efficiencies (clear-sky aerosol-free ERF$_{csaf}$ for UKESM) for *2xNOX* lightning NOx emissions. Tropospheric and stratospheric ozone column changes and their estimated radiative effects. Changes in emission from *4xCO2* experiment. α values are calculated assuming ERF$_{csaf}$ or a SARF efficiency for ozone of 0.042 W m$^{-2}$ DU$^{-1}$. Uncertainties for each model are errors in the mean based on interannual variability, and assuming a 14% uncertainty in the ozone radiative efficiency. Uncertainties in the multi-model results are standard deviation across the models.**


As with BVOC emissions (above) the potential impacts of lightning on methane lifetime can be diagnosed. All models show (table 13) a decrease in methane lifetime with increased lightning NOx emission from -2.3 to -4.8 % (Tg yr$^{-1}$)$^{-1}$. The feedbacks are negative for UKESM1 and CESM2-WACCM (0.007 and 0.012 W m$^{-2}$ K$^{-1}$) and slightly positive for GFDL-ESM4 (0.001 W m$^{-2}$ K$^{-1}$) and almost exactly cancel out the feedback due to the ozone column. The net (ozone + $\tau_{CH_4}$) feedbacks for

UKESM1, GFDL-ESM4 and CESM2-WACCM are -0.002, 0.000, and 0.001 W m$^{-2}$ K$^{-1}$. For UKESM1 a feedback of -0.006 W m$^{-2}$ K$^{-1}$ should be added to account for the increase in sulphate.





| | UKESM1 | GFDL-ESM4 | CESM2-WACCM | Multi-model |
|---|---|---|---|---|
| $\tau_{CH_4}$ /emission % $(Tg\ yr^{-1})^{-1}$ | -2.3 | -3.8 | -4.8 | |
| $\tau_{CH_4}/\Delta T$ % $K^{-1}$ | -0.6 | 0.1 | 1.0 | |
| $\alpha\ \tau_{CH_4}$ $W\ m^{-2}\ K^{-1}$ | -0.007±0.001 | 0.0012±0.0004 | -0.012±0.002 | -0.006±0.005 |

**Table 13. Percentage change in methane lifetime for lightning NOx emissions. Estimated change in lifetime following changes in NO$_X$ emission from *4xCO2* experiment. α values are calculated assuming a radiative efficiency of 0.015 W m$^{-2}$ %$^{-1}$.  Uncertainties for each model assume a 14% uncertainty in the methane radiative efficiency. Uncertainties in the multi-model results are standard deviation across the models.**

### 4.2.3 Wetland emissions

Although the wetland emissions do not directly affect methane concentrations in the model, changes in emission can be converted to concentration changes (section 2.2). UKESM and WACCM models with interactive wetland emissions show strong responses to climate change, leading to a feedback of 0.16±0.03 W m$^{-2}$ K$^{-1}$.

| | UKESM1 | CESM2-WACCM | Multi-model |
|---|---|---|---|
| 4xCO2 Tg yr$^{-1}$ K$^{-1}$ | 40 | 60 | |
| $\alpha$ W m$^{-2}$ K$^{-1}$ | 0.13±0.02 | 0.19±0.03 | 0.16±0.03 |

**Table 14. Sensitivity of wetland emissions to *4xCO2* in two models. Feedback parameter assuming pre-industrial conditions. Uncertainties for each model assume a 14% uncertainty in the methane radiative efficiency. Uncertainties in the multi-model results are standard deviation across the models.**

### 4.2.4 Temperature and humidity

As well as through changes in natural emissions, climate change can affect ozone burden and methane lifetime directly as the production and loss reactions are sensitive to temperature and water vapour. Here we add the expected changes in ozone and methane lifetime due to changes in BVOCs and lightning NO$_X$ from sections 4.2.1 and 4.2.2 above and compare those to the changes diagnosed from the 4xCO2 experiments (table 15). The residual is then the direct effect of climate. For CESM2-WACCM and GFDL-ESM4 most of the total increase in tropospheric ozone can be explained by the changes in natural emissions (particularly BVOC) suggesting that non-emission drivers of tropospheric ozone change (temperature, humidity,





transport from the stratosphere, dry deposition) balance to have little net effect. Increases in stratospheric ozone are much

larger than expected from the changes in natural emissions, suggesting that meteorological changes (principally cooling stratospheric temperatures) are the main driver. The tropospheric ozone change attributable to climate can be used to determine a feedback which is only significant for UKESM1 (-0.023 W m$^{-2}$ K$^{-1}$). The stratospheric ozone changes cannot simply be converted to an ERF, since unlike for the natural emission (where the ozone changes were close to the tropopause) the tropospheric radiative efficiency cannot be applied.

In UKESM and GFDL-ESM4 the meteorological changes decrease methane lifetime by similar amounts (-4.5 and -4.6 % K$^{-1}$) and hence have similar feedbacks (-0.078 and -0.080 W m$^{-2}$ K$^{-1}$). In the case of GFDL-ESM4 this leads to an overall decrease in lifetime rather than the increase expected from natural emission changes (principally BVOC). In WACCM the overall effect of climate is to increase the methane lifetime, almost entirely due to the increased BVOC emissions with little effect of meteorological drivers. This is surprising since there is no known mechanism whereby temperature and humidity increases

can increase the methane lifetime. This could be due to non-linearity whereby the effect of increased VOCs on methane lifetime is larger than expected from scaling the *2xVOC* experiment.

Combining the results from ozone and methane lifetime changes leads to overall feedbacks from temperature of -0.101, -0.082 and +0.015 W m$^{-2}$ K$^{-1}$ for the three models.




| | UKESM1 | | GFDL-ESM4 | | CESM2-WACCM | | Multi-model |
|---|---|---|---|---|---|---|---|
| | trop | strat | Trop | strat | trop | strat | |
| LNOx+BVOC Ozone (DU K$^{-1}$) | 0.210± 0.007 | 0.102± 0.002 | 0.160± 0.007 | 0.247± 0.006 | 0.151± 0.005 | 0.734± 0.008 | |
| $4xCO2$ Ozone (DU K$^{-1}$) | -0.33±0.02 | 1.28±0.10 | 0.18±0.02 | 3.46±0.06 | 0.162±0.003 | 2.82±0.04 | |
| Ozone residual (DU K$^{-1}$) | -0.54±0.02 | 1.18±0.10 | 0.02±0.02 | 3.22±0.06 | 0.011±0.006 | 2.09±0.05 | |
| α Ozone residual W m$^{-2}$ K$^{-1}$ | -0.023±0.003 | | 0.001±0.001 | | 0.000±0.000 | | -0.007±0.011 |
| LNOx+BVOC $\tau_{CH_4}$ % K$^{-1}$ | 0.43±0.04 | | +2.55±0.07 | | +6.05±0.09 | | |
| $4xCO2$ $\tau_{CH_4}$ % K$^{-1}$ | -4.08±0.02 | | -2.05±0.06 | | +6.92±0.06 | | |
| $\tau_{CH_4}$ residual % K$^{-1}$ | -4.51±0.04 | | -4.60±0.08 | | +0.87±0.11 | | |
| alpha $\tau_{CH_4}$ residual W m$^{-2}$ K$^{-1}$ | -0.078±0.011 | | -0.080±0.011 | | 0.015±0.003 | | -0.048±0.045 |

**Table 15. Comparison of expected changes in ozone column and methane lifetime with that diagnosed from *4xCO2*. Residual is given by the difference and is converted to a feedback using radiative efficiencies for tropospheric ozone and methane lifetime.**

### 4.3 Overall feedback

The feedbacks are summarised in table 16. The largest individual feedbacks are due to the generation of aerosols by BVOCs (-0.113±0.102 W m$^2$ K$^1$) and the emission of methane from wetlands (0.16±0.03 W m$^2$ K$^1$). The overall uncertainty is dominated by the uncertainty in the aerosol response to BVOC emissions. Nearly all the feedbacks are negative, most because they come from an increase in aerosol with temperature. For BVOC emissions, the increase in aerosols outweighs the increases in ozone and methane. For lightning NOx, the ozone and methane changes cancel. A warmer and more humid climate also leads to less ozone and methane.

The ESMs that use the *abrupt-4xCO2* experiment to quantify the climate sensitivity do not allow methane to vary, so we also quantify the non-methane feedbacks that will be contributing to the diagnosed climate sensitivity in these models. This





feedback is significantly negative (-0.228±0.123 W m$^{-2}$ K$^{-1}$) suggesting the climate sensitivity of ESMs might be expected to be lower than for their physical-only counterparts.

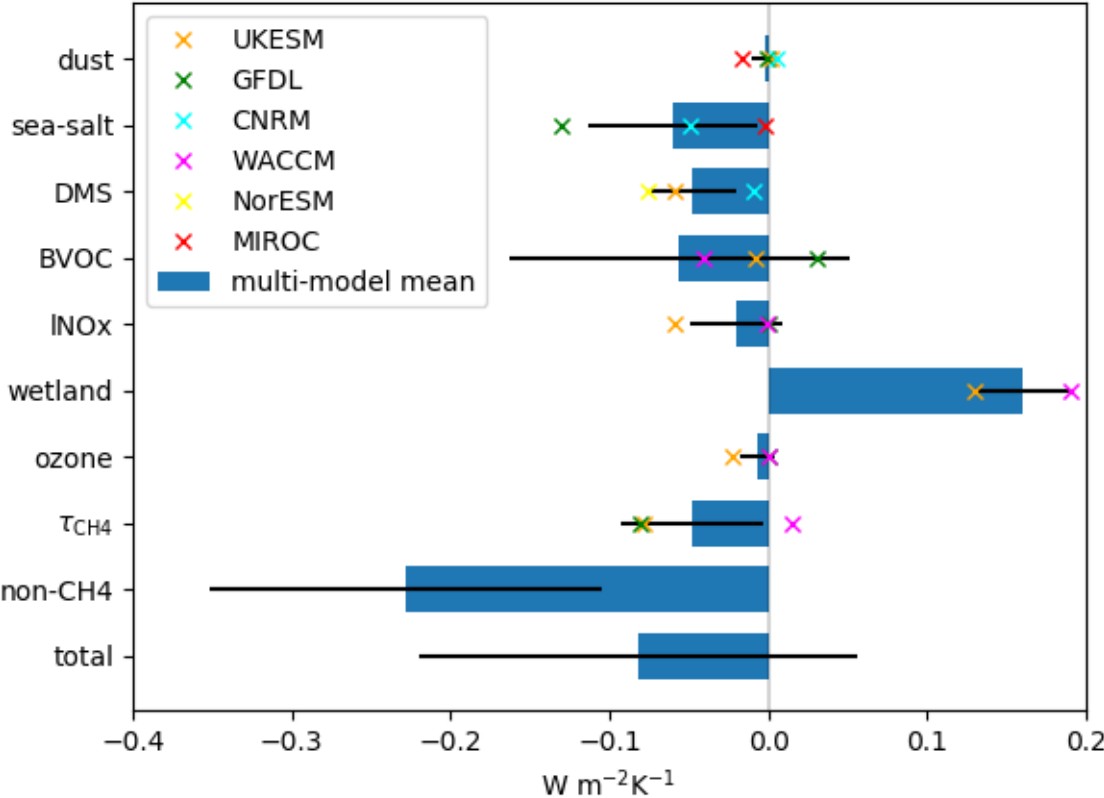


**Figure 5. Feedback parameters of all the aerosol and chemical processes in table 16. Multi-model mean and individual models. Uncertainties are inter-model standard deviations. BVOC and lNOx include aerosol, ozone and methane lifetime effects (points are only shown for models that include all effects). Non-CH4 excludes methane lifetime effects and wetland feedback.**





| Process | Feedback parameter $\alpha$ (Wm$^{-2}$ K$^{-1}$) |
|---|---|
| Dust (AOD) | -0.003±0.008 |
| Sea Salt (AOD) | -0.060±0.053 |
| DMS and sulphate lifetime | -0.048±0.028 |
| BVOC (Aerosol mass) | -0.113±0.102 |
| BVOC (ozone) | 0.017±0.011 |
| BVOC ($\tau_{CH_4}$) | 0.041±0.030 |
| lNO$_X$ (Aerosol) | -0.002±0.003 |
| lNOx (ozone) | 0.006±0.004 |
| lNOx ($\tau_{CH_4}$) | -0.006±0.005 |
| Wetland | 0.163±0.032 |
| Chemistry (ozone) | -0.007±0.011 |
| Chemistry($\tau_{CH_4}$) | -0.048±0.045 |
| Total non-methane | -0.228±0.123 |
| Total | -0.082±0.138 |

**Table 16. Feedback parameters of all the aerosol and chemical processes addressed in this study. Uncertainties are inter-model standard deviations.**

## 5. Discussion

### 5.1 Dust

Dust-aerosol feedback assessments are a relatively new area of research owing to the large uncertainties of climate models in simulating dust aerosols with changes in atmospheric composition. For instance, the spread in model estimates for dust aerosol changes in the 21st century is the largest among wildfires, biogenic SOA and DMS sulphate (Carslaw et al., 2010). Predictions for future dust emission range from an increase (Woodward et al., 2005) to a decrease (Mahowald and Luo, 2003). The modelled feedbacks in section 4.1.1 have a range of -0.016 to +0.048 W m$^{-2}$ K$^{-1}$ compared to the theoretical model estimates of −0.04 to +0.02 Wm$^{-2}$K$^{-1}$ by Kok et al. (2018).

The model ranges in dust forcing and feedbacks are not surprising in light of past studies that highlight model differences in dust-emitting winds and dust-aerosol parameterizations that contribute to the model diversity in the dust-aerosol loading, optical properties, and radiative effects (Ackerley et al., 2012; Evan et al., 2014; Huneeus et al., 2011; Shao et al., 2011; Zender et al., 2004). For instance, the parameterization of the planetary boundary layer plays an important role in determining the dust





loading, forcing, and regional feedbacks on winds (Alizadeh Choobari et al., 2012). Influencing factors for regional differences in the dust radiative effects are the surface albedo and aerosol size distribution (Kok et al., 2018; Xie et al., 2018), whereas

feedbacks on winds depend also on meteorological factors (Heinold et al., 2008). The substantial model differences in the dust emission response to *4xCO2* paired with corresponding differences in mean 10m-wind speed in this study suggests that also the dust feedback parameter critically relies on accurately simulating atmospheric dynamics. Modelling atmospheric circulation has been identified as a grand challenge in climate research (Bony et al., 2015). Currently, we have no estimate which of the dust feedbacks shown are the most plausible, because convective dust storms are missing in such models, but this

dust storm type is believed to be important for North African dust emissions (Heinold et al., 2013). Moreover, natural aerosol-climate feedbacks are thought to depend on the anthropogenic aerosol burden and might therefore be both time-dependent and underestimated in the present-day polluted atmosphere (Spracklen and Rap, 2013). Taken together, we have a low confidence in the feedback estimates for dust aerosols to increases in atmospheric concentrations of greenhouse gases.


## 5.2   Sea Salt

The doubled sea salt ERF in section 4.1.2 is -0.35 to -2.28 W m$^{-2}$, higher end than found in the literature (-0.3 to -1.1 W m$^{-2}$ which is for direct forcing only (Yue and Liao, 2012)). The efficiency per AOD ranges from -20 to 39 W m$^{-2}$, again higher

than the literature for direct forcing (-18 to -24 W m$^{-2}$ (Heald et al., 2014; Yue and Liao, 2012)).

## 5.3 DMS

DMS is produced by marine biological activity in the ocean, and it is assumed to be the largest natural source of sulphur to the atmosphere.  Up to now, there has been no comprehensive model effort to include all the important effects, and therefore the DMS emission strength change under climate change is still uncertain. The range here (-0.010 to -0.075 W m$^{-2}$ K$^{-1}$ including

increases in sulphur lifetime) encompasses  the -0.02 W m$^{-2}$ K$^{-1}$  from AR5 (Ciais et al., 2013), based on results from only one model (HadGEM2-ES).

DMS production is closely linked to primary production.  Modelling studies including ocean biogeochemistry have shown that under climate change, an increased stratification of the ocean at low and mid latitudes leads to a reduction in nutrients supply

into the surface ocean and thus a reduction in DMS emissions, whereas at high latitudes, retreat of sea-ice can lead to increased primary production and increase in DMS production (Kloster et al., 2007).  Globally, most models which include ocean biogeochemistry show a slight increase in DMS production and emission to the atmosphere in a warming climate (Bopp et al., 2004; Gabric et al., 2004; Gunson et al., 2006; Vallina et al., 2007).


Some more recent studies have included the impact of ocean acidification on ocean DMS production (Schwinger et al., 2017; Six et al., 2013). Both studies used a very similar description of the ocean biogeochemistry and extended it with an observationally-based relation between ocean alkalinity and ocean DMS production. Assuming a medium sensitivity of the DMS production on pH, Six et al. (2013) found a global DMS emission decrease by 18% in 2100 under the SRES A1B scenario, and Schwinger et al. (2017) an emission reduction by 31% in 2200 under the RCP8.5 scenario. In addition recent

work has provided evidence for major pathways in the oxidation of DMS in the atmosphere which are not included in any of these ESMs ((Berndt et al., 2019; Wu et al., 2015).

### 5.4 BVOC

When emissions of BVOCs are increased we see changes to organic aerosol concentration and (in some models) the atmospheric concentrations or lifetime of $O_3$ and $CH_4$, with competing effects on climate. At the multi-model mean level, the

cooling associated with an increase in organic aerosol ($-0.113\pm0.102$ W m$^{-2}$ K$^{-1}$) outweighs the warming associated with an increase in $O_3$ ($0.017\pm0.011$ W m$^{-2}$ K$^{-1}$) and an increase in $CH_4$ lifetime ($0.041\pm0.030$ W m$^{-2}$ K$^{-1}$).

Using multi-annual simulations of global aerosol, Scott et al. (2018) diagnosed a feedback from biogenic secondary organic aerosol of $-0.06$ W m$^{-2}$ K$^{-1}$ globally, and $-0.03$ W m$^{-2}$ K$^{-1}$ when considering only extra-tropical regions. This global feedback value was composed of a direct aerosol radiative feedback of $-0.048$ W m$^{-2}$ K$^{-1}$ and an indirect aerosol (i.e., cloud albedo)

feedback of $-0.013$ W m$^{-2}$ K$^{-1}$. Using observations from eleven sites, Paasonen et al., (2013) estimated an indirect aerosol feedback of $-0.01$ W m$^{-2}$ K$^{-1}$ due to biogenic secondary organic aerosol. The ability of models to account for changes in vegetation has a significant effect on the feedback. Sporre et al (2019) found interactive vegetation, enhanced BVOC emissions by 63% greater relative to prescribed vegetation, producing more organic aerosol and an increase in (negative) aerosol forcing. The level of compensation between increased aerosol forcing and increased ozone and methane lifetime is dependent on the

model (here positive feedback for GFDL, negative for UKESM1 and WACCM). Unger (2014) found a positive feedback in NASA GISS ModelE2, whereas Scott et al. (2014) found a negative feedback in HadGEM2-ES.

### 5.5 Lightning

The ESMs used in CMIP6 all use a cloud-top height parameterisation of lightning. Such schemes have previously been found to increase lightning production in warmer climates whereas more sophisticated schemes based on convective updraft mass

flux or ice flux show decreases in lightning with temperature. (Clark et al., 2017; Finney et al., 2016b, 2018). Two models here (UKESM1 and WACCM) show increases in lightning emissions of 0.27 and 0.21 Tg(N) yr$^{-1}$ K$^{-1}$ which is slightly lower than the results from the Atmospheric Chemistry and Climate Model Intercomparison (ACCMIP) of 0.44 Tg(N) yr$^{-1}$ K$^{-1}$ (Finney et al., 2016a).





### 5.6 Wetland methane

Wetland emissions are more strongly sensitive to $CO_2$ concentrations than to temperature or precipitation (Melton et al., 2013), so the values presented here are more likely to be "adjustments" to the $CO_2$ rather than feedbacks, and hence could be considered part of the $CO_2$ ERF. We find emission increases following quadrupled levels of $CO_2$ of 130-160%. This compares with results from the Wetland CH4 Inter-comparison of Models Project (WETCHIMP) of 20-160% following an increase in $CO_2$ of a factor of 2.8 (Melton et al., 2013). The CMIP6 simulation specifications do not include free-running methane

concentrations therefore the effects of these increased wetland emissions will not be realised in any of the CMIP6 experiments. Outside CMIP6, ESMs are starting to include free-running methane (Ocko et al., 2018), so for these it will be important to understand the effects of changing $CO_2$ and meteorology on wetland emissions.

### 5.7 Temperature and humidity discussion

We find a decrease in methane lifetime of -4.5 to -4.6 % $K^{-1}$ in UKESM1 and GFDL-ESM4, but an increase of 0.9 % $K^{-1}$ in

WACCM. The first two models compare well with ACCMIP which found a sensitivity of 3.4±1.4% $K^{-1}$ (Naik et al., 2013; Voulgarakis et al., 2013). The impact of climate (including natural emission changes) on tropospheric ozone varies from negative in UKESM1 ( -0.33 DU $K^{-1}$) to positive in GFDL-ESM4 and WACCM (0.18 and 0.16 DU $K^{-1}$). ACCMIP also found a variation in sign amongst models −0.024±0.027 W $m^{-2}$ for a 1850-2000 change in climate (equivalent to -0.57±0.64 DU using the same radiative efficiency as table 15). The ACCMIP models generally did not include stratospheric chemistry so

either explicitly prescribed the cross-tropopause flux of ozone or imposed a climatology of ozone above the tropopause. The three CMIP6 models here all treat the chemistry seamlessly across the troposphere and stratosphere so the impact of changes in stratosphere-troposphere exchange (STE) of ozone on the tropospheric column is likely to be different to ACCMIP.

Changes in the stratospheric ozone column following a quadrupling of $CO_2$ are driven by cooling temperatures in the stratosphere. This is likely to be due to temperature adjustments to the stratospheric $CO_2$ concentrations, and so part of the

ERF for $CO_2$ rather than a feedback (Smith, submitted). Feedbacks and adjustments cannot be distinguished with this experimental setup.

### 6 Conclusions

Earth system models include more processes than physical-only climate models. These models will inherently include additional climate feedbacks, and so have a different overall climate feedback (and climate sensitivity) to their physical

counterparts. In this study we consider six earth system models (CNRM-ESM2, UKESM1, MIROC6, NorESM2, GFDL-ESM4, and CESM2-WACCM). Five of these (CNRM-ESM2, UKESM1, MIROC6, NorESM2 and GFDL-ESM4) participated in the aerosol-related feedback experiments, and three (UKESM1, GFDL-ESM4, and CESM2-WACCM) in the chemistry-related feedback experiments.



We focus in this study on the responses to an abrupt forcing of quadrupled $CO_2$ concentrations as that is the usual method to

diagnose climate feedbacks. By convention the feedbacks are quantified as a response to temperature (in W m$^{-2}$ K$^{-1}$), but they may not necessarily be applicable to drivers of climate change other than $CO_2$ as some of the "feedbacks" may be instead adjustments to $CO_2$ concentrations. It should also be noted that *abrupt-4xCO2* feedbacks are based on atmospheric conditions representative of 1850s and thus may not be applicable to future responses starting from present day conditions.

Here we find that the dominant feedbacks are negative i.e. that they act to dampen the response to an imposed forcing. The

total feedback, excluding inferred changes in methane, is -0.228±0.123 Wm$^{-2}$ K$^{-1}$. The increase in organic aerosols from increase emission of volatile organic compounds (VOCs) from vegetation makes the largest contribution to both the magnitude of the feedback and its uncertainty (-0.113±0.102 Wm$^{-2}$ K$^{-1}$) with increases in sea salt and sulphate aerosol also contributing. The increase in sulphate comes both from an increase in DMS emissions and a decrease in sulphate removal.

Contributions from increases in ozone production from biogenic VOCs and lightning NOx are partially offset by decreased

tropospheric ozone lifetime in a warmer climate. Stratospheric ozone does substantially increase. Diagnoses of changes in wetland emissions of methane indicate that if ESMs did allow methane to vary interactively the combined aerosol and chemical feedbacks would be substantially less negative and consistent with zero.

**Acknowledgements**

GT, WC, RC-G, MM, FO'C, DO, MS, CES acknowledge funding received from the European Union's Horizon 2020 research and innovation programme under grant agreement No 641816 (CRESCENDO). MM acknowledges H2020 CONSTRAIN under the grant agreement No 820829. CES acknowledges funding from the Natural Environment Research Council (NE/S015396/1). FO.C, GF and JPM were supported by the Met Office Hadley Centre Climate Programme funded by BEIS and Defra (GA01101).


**Author Contributions**
Manuscript preparation was by WC, GT, DO, RC-G, CES, SF and additional contributions from all co-authors. Model simulations were provided by SB, GF, AG, , J-FL, MM, JM, PN, TT. Analysis was carried out by GT, WC, DO, SF, RC-G, JW.



**Data Availability**

All data from the Earth system models used in this paper are available on the Earth System Grid Federation Website, and can be downloaded from there. https://esgf-index1.ceda.ac.uk/search/cmip6-ceda/

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
