# Peer review of "Climate-driven chemistry and aerosol feedbacks in CMIP6 Earth system models"

_Atmospheric Chemistry and Physics, 2019_

## Referee Comment (RC1) · Anonymous Referee #1 · 8 Apr 2020

The paper is an interesting summary of the magnitude of chemistry and aerosol feedbacks in available CMIP6 climate models. The paper is generally well-written, however in its current form the manuscript is somewhat fragmented and some important discussion about is missing. Some aspects of the methodology are described concisely, yet some important details are missing entirely, or are described only briefly. The chemical and aerosol forcing agents are considered independently which helps compartmentalise the results and some of these section include important insights. However, other sections have not been crafted with the same care.

The paper would benefit from merging sections 4 and 5. Currently results from several CMIP6 climate models are somewhat mechanistically portrayed in section 4. Section 5 contains some context for interpreting the differences between models, but uses identi-

cal subsection headings and much of the content is more suited to the introduction of a paper on one or more of the forcing agents. The chosen format makes the manuscript unnecessarily disjointed and does not help contextualise the main results. Once sections 4 and 5 are merged, they should be revised to include discussion of the physical processes that cause differences between models. Currently, this is only achieved for one or two of the forcing agents.

The article has two main themes. Firstly, the differences in aerosol and chemistry forcing efficiency and burden sensitivity are considered. Secondly, the magnitude of feedbacks from forcing agents are contrasted. It is not clear what the authors intended the main message of the paper to be. The abstract provides very few conclusions about either of these aspects and is overly focussed on methane-specific results. If the paper is intended to focus on the second aspect, then the majority of the feedback summary tables could be moved to the SI without reducing the impact of the paper. However, I think it would be better to retain these tables and include a process-based discussion of the causes of model differences as suggested above.

The use of standard deviations to represent uncertainty in a handful of models is not appropriate. It is possible that this is not what the authors have done, but their method is currently unclear. The authors need to clarify their multi-model uncertainty calculations in the text and if they are currently using standard deviations to represent uncertainty in only 3-6 values, need to seek more appropriate ways to communicate this information. Currently multi-model uncertainties are communicated through table captions, but should be described fully in the main text.

Some other points that require attention include:

All figures require subfigure labels as per ACP guidelines, to match references in the captions and main text.

Line 34: "with warmer temperatures" needs a fuller description. 4Xco2 induced warming

[Figure]

Line 37: VOC needs to be defined.

Line 40: GCMs do these things already. ESMs include the interactions between these systems, by coupling them and hence can expect a greater degree of consistency of information across model components. This needs to be clarified in the text.

Line 57: Here and in the conclusions, it is important to mention that some of the forcing agents considered make important climate contributions at the regions scale that are neglected when global mean temperatures are used to represent climate change.

Line 102: The scale factor is not well justified. The cited document is a substantial IPCC chapter. Presumably, authors are referring to section 8.2.3.3? Including the page number would help reader. However, the derivation of the scale factor used here is unclear and some explanatory text is required.

Line 105: The use of the value 9.25 also needs justification and a description of how it corresponds to values supplied in the referenced document.

Line 110: "four have . . . and three have . . ." is ambiguous. "Three of these four also have. . ." is clearer. Table 2 makes this clear, but is not currently referenced.

Line 119: Table 3 is currently referenced in a way that suggests it will compare emissions from all natural sources, whereas it actually shows differences between models for dust and BVOCs only. The text needs to be revised. This error is repeated on the first line of section 3.2.2.

Table 3: "PAR" needs to be defined. The phrase "Not dependent on vegetation" is redundant.

Table 4: There are inconsistencies in the table. For example, sometimes "wind" is used and at other times "wind speed dependent". Descriptions here are too brief. What is the difference between DMS emission and oceanic organic aerosol complexity for NorESM2-LM and UKESM1 for example?

Section 4, Line 150: Section 2.1 should be referenced in the first paragraph, so that the normalization of temperatures can be put in the context of $\gamma i$ as defined in that section.

Line 155: For non-specialist readers an indication of the number of years required to reach equilibrium on average is needed.

Line 163: Figure S1 does not obviously support this claim. Global mean ERF values should be provided for each model. Also, the authors should explicitly state they are discussing "global mean" effective radiative forcing here.

Line 165: The strong regional forcing over Africa should be mentioned as the primary cause of positive SW forcing. Some speculation of the process parameterisations that cause this model behavior should be given.

Line 182: Refer to table 6 again. Also, some speculation on the physical processes causing the increased lifetime should be given. This is a good example of the need for additional discussion and how merging, then adapting content from section 5 will improve the interpretation of results. Line 189: It is not clear what the 2nd use of "for instance" here is referring to. This sentence needs to be rewritten to improve clarity.

Table 6: The reason for missing values in this and other tables needs to be explained more clearly within the text.

Line 200: These forcing values are far larger than for dust. Are the forcing-emission-feedback relationships expected to be linear? If not, there will be discrepancies in the gamma terms across emission types, even if normalised. This assumption on linearity and its implications need to be discussed here and/or in section 2.2.

Line 201: Why 20x? Is this caused by the choice of size bins? This warrants some discussion. Why is the AOD of a similar magnitude? What model processes have been adjusted/tuned to make the AOD similar? The reasons models have similar values for very different reasons need to be better understood. This is important for understanding the causes of model diversity in climate projections.

Line 205 - 209: All positive except MIROC needs to be explained/considered. What regions show a decrease in emissions that causes the global mean response to be negative? Maps for each model in the SI are needed.

Line 220: It should be explained here that all models could have run the 2xdms experiment. Interactive ocean biogeochemistry is not a prerequisite, since emissions could have been scaled within the flux parameterisation as with the 2xdust experiment.

Line 222: Fig 3 does not show the forcing values for each model as implied. Table 8 should be referenced to here.

Line 222-224: Maps of sulphur concentrations and changes in concentrations need to be included as a figure in the SI for each model, so the reader has a clear understanding of the magnitude of regional compensation across models.

Line 225: GFDL-ESM4 values only contributes to the multi-model sensitivity to emissions/concentrations, but not to the multi-model radiative efficiency. The assumption made here is that all models have similar radiative efficiencies. This is an important assumption, given the diversity of model responses highlighted up to this point in the manuscript. Is it appropriate to assume GFDL-ESM4 has the same radiative efficiency as the two models used in the sensitivity calculation? Some justification is required if the authors want to maintain this approach. An alternative approach would be to only use the 2 models with sufficient information to calculate both the multi-model sensitivity and multi-model radiative efficiency. This subjective choice to include partial information from one model needs to be justified more clearly and the implications of extrapolating the multi-model radiative efficiency to other models needs to be considered and openly discussed.

Line 229: The magnitude of the increase should be quantified in the text. Line 249: Here and elsewhere in the text, the word "significant" is used without mention of associated statistical tests. The values should be state with "significant" removed, or the methodology more accurately described.

Line 253: Incorrect label. Figure S3 only shows the multi-model mean. Given the diversity in aerosol forcing from this source, maps of CDNC should be provided for each model. Also, interpretation of the differences between models needs to be included here.

Line 253-257: Examples of regions where these behaviors are likely, with an explanation of why is needed.

Figure 4: Fig S3 could be a subfigure of Fig 4.

Table 9: Uncertainty values are missing for UKESM1 and multi-model mean values are missing for Scaled Mass

Line 278: Is there an hypothesis the authors could provide to explain the causes of model diversity in BVOC partitioning into ozone and aerosol forcing? This sort of discussion is essential to develop a better understanding of the importance model differences and will affect interpretation of climate feedbacks across models.

Table 10: There is no explanation of why 14% is used. This should be in the methods section, not hidden in a caption.

Line 300-302: This sentence needs to be rewritten to improve readability.

Line 302: It is not clear from the text as written, how BVOC burden sensitivities are used in the methane sensitivity calculation.

Section 4.2.4: The title of this section is misleading. Several non-emission drivers are considered, not just these two.

Line 265: "1" missing from UKESM1.

Section 4.3: This section needs some comment about the importance of climate forcing agents that have climatic importance at the regional scale, to prevent the results of this manuscript being interpreted incorrectly.

[Figure]

Line 380: The authors need to specify that these are multi-model feedbacks, here and in the table caption. Figure 5 needs to be referenced. In addition, the cancellation between models with opposite signs again needs to be mentioned within this section, as does the fact that a different number of models were used to calculate the multi-model means because of data availability.

Line 402: Can the feedbacks be interpreted in the context of the magnitude of forcing from these forcing agents over some specified period? Uncertainty in these magnitudes should be included in the discussion with appropriate references.

Line 423: There is no use citing these values if not directly comparable. This text should be removed to avoid confusion. Further discussion of the causes of model differences is required here.

Line 433: Please clarify the difference between primary production and DMS production in the text.

Line 505: This value needs context to aid interpretation. e.g. What is this as a proportion of the GHG forcing required to increase temperatures by 1 degree?

Line 507-508: The uncertainties in these values are substantial and need to be included in this discussion and interpretation of results.

SI: S1, some descriptions are missing entirely and need to be included.

SI: All figures require subfigure labels.

———————————————————————

---

## Referee Comment (RC2) · Anonymous Referee #2 · 24 Apr 2020

Thornhill et al. analyse a set of Earth System Model simulations with atmospheric chemistry and aerosol parameterisations to quantify climate feedbacks associated with aerosol and chemistry processes. The methodology allows to attribute the climate feedback to different chemical and aerosol processes and thereby provides in some cases important insights. The paper is highly relevant and fits well to the scope of ACP. The paper is generally well written, but the quality of the individual sections varies considerably.

The abstract should list the feedbacks assessed here and should be much more explicit about the major findings of this study (I would assume that this would be a summary of Figure 5). It is unclear to me why the methane effects are highlighted here, while this is not mentioned at all in the Conclusion section. In general, the authors should try to

clarify the main messages from this paper in abstract, introduction and conclusions.

The introduction is somewhat simplistic in that it only lists studies that have attempted to assess non CO2 climate feedbacks. For the general audience and the orientation of the readers it would be helpful to start with a somewhat more detailed description of the major processes and feedbacks considered here and why they matter to the climate system.

The choice of the authors to rely on 4xCO2 experiments to diagnose climate feedback implies that some of the feedbacks considered are less climate change related, but mediated by the effect of CO2 on vegetation productivity and cover. This is an important caveat that should be explained in the Methods section for those processes that do respond to CO2 as well as climate. Also, this needs to be reflected critically in the Conclusions section/Abstract.

The methods section should be expanded by a description on how the authors have dealt with uncertainty in this study. What do the reported $\pm$ ranges represent for individual estimates, how are errors of the multi-model mean derived from these (error propagation of the IAV?), how is the error range of the total forcing estimate determined, how have varying estimates from emission/burden based methods been dealt with in the total feedback assessment.

I rarely recommend merging results and discussion, but I agree with reviewer #1 that in this case, where a lot of different processes are at play, it would be advisable to merge section 4 and 5 in the sense to have results and discussion for each of the different forcing agents together. The quality of the results presentation and their discussion varies substanially, and the authors should strive to be more explicit in terms of describing and explaining the important differences between models, and where possible provide an appropriate comparison to previous studies. There should then still be an overall discussion section 5/6 in the end where the overall contribution of the non-CO2 chemistry and aerosol feedbacks are discussed in the light of other climate feedbacks

(physical, carbon, ...).

Minor comments:

L35: define warmer temperatures

L36: positive methane feedbacks?

L44: consider adding Arneth et al. 2010, Nat. Geo (Doi: 10.1038/ngeo905) to this list

L72: Briefly explain why you not just use one of the options. I also think that this question deserves more attention in the results section where you for some forcings can compare the magnitude of the alternative estimates more systematically to derive at a joint assessment of the individual feedback factor.

Section 2.2: It would be helpful to know which of the feedbacks is calculated which way here. Also, given the need for standardisation here or in the discussion section, there should be a discussion about the assumption of linearity of the radiative forcing response to emissions/burden across a large range of emissions/burden.

Which ensemble members were selected for this study, or does the study use an ensemble mean?

L86: It is unclear whether this is based on simulations presented in Collins et al. 2017, or based on new AerChemMIP experiments, please clarify

L102: Provide an explanation for this scaling factor rather than referencing a full IPCC-chapter

L105: For completeness, give value assumed for M_atm as well as the molecular masses of CH4 and air

Section 3.1 should reference table 2 but does not.

Section 3.2: natural emissions of what?

L119: this sentence needs to be clarified. There are multiple climate-relevant landbased emissions beyond dust and BVOCs. What do the authors want to state here?

Table 3: define LAI, PAR. Given an indication what LAI varies and interactive vegetation imply. The table captions says BVOC, the header VOC, which is correct?

Given that Section 4.2.3. discusses wetland emissions, the models used should be described here briefly as well.

L134: same as L119

Table 4: what is the difference between wind dependent and wind speed?

L147: Does this sentence imply all models use the same paramterisation?

L151: refer back to Section 2.1 or remove as this is partly redundant.

L155: For the non-expert reader, explain how long the development of a new equilibrium takes and how large the difference on average would be.

Table 5: No SD?

L163: Figure S1 does not separate shortwave and longwave effects to make this claim.

L165: the positive shortwave forcing OF DUST AEROSOLS? Is it possible to provide an explanation for this CNRM response?

Figure 1 (and subsequent following figures): use stippling or alike to show areas of model dis-/agreement. Also revise figures to ensure the legend is readable without magnifying glasses

Table 6 (and similar subsequent tables): Why are certain cells blank?

L201: Why does this discrepancy occur, and how can the AOD be still similar? This paragraph should also have a discussion on why MIROC6 deviates in terms of the ERF response

L222: Figure 3 does not show this.

Table 8: Please check values, at least the alpha emission multi model mean cannot be correct.

L253: Figure S4 does not exist.

L279: BVOC-related aerosols, or aerosols in general?

L281: refer back to Section 2.2.

L297: Methane Burden/Emissions? does not change

L304: The 0.015 Wm-2 %-1 are not described in Section 2.2. but should be

Section 4.2 general: I think it would be easier to follow if the indirect effects of NOx and BVOC on methane were discussed jointly and possible even in one table, as they rely on the same methodology and type of experiments

Section 4.2.3 I find this section troublesome given the lack of explanation of the simulated methane emissions, particular because this presentation confounds the direct effects of CO2 on methane emissions (via CO2 fertilisation of wetlands) with the direct effects of temperature on methane-emissions, but exclusively attributes this to temperature. The result of which is an inflated methane-emission climate feedback compared to Ciais et al. 2013. I wonder whether there are simulations with interactive methane but no biogeochemical coupling to CO2 available from the C4MIP project that would allow to tackle this separation? As a minimum, this confounding effect needs to be explained and discussed.

Table 14: What is the justification to assume at 14% uncertainty on methane radiative efficiency?

Section 4.2.4 should be labelled atmospheric temperature and water vapour?

L356: the residual is then ASSUMED TO BE the direct effect. This statement could be backed up by a brief explanation that BVOC and NOx are the only agents affecting ozone and methane lifetimes next to climate in these models. Otherwise, it should be

explained why other factors may be small and negligible.

L367: Consistently use CESM-WACCM

Section 4.3: Figure 5 is not referenced. The text needs to be explicit that the feedbacks are the multi-model mean, and that not all feedbacks could be calculated for all processes considered. A discussion that I have been missing here is whether these terms are really additive and linear as assumed. It is possible that there is a compensation of feedbacks between models, so I wonder whether it would be possible / interesting to compare the sum of feedbacks across processes for those models that have calculated similar feedbacks

Figure 5: use consistent labelling of models. use consistent labelling of forcing factors (e.g. total non-CH4, wetland CH4 etc.) Use a clearer abbreviation for lightning NOx than lNOx. The figure caption should also explain, how and why feedbacks from table 16 were aggregated in the figure.

Section 5.2 is not helpful is no guidance is given as to the origin of the large range in the estimates and the plausiblity of the different model projections. The comparison to the literature numbers is insufficient in that the numbers aren't directly comparable. This section needs substanial revision.

Section 5.6 response to my previous comment, but then implies that this shouldn't really be listed here as a climate feedback, but a biogeochemical carbon-methane feedback.

Section 6: I would have liked to see a somewhat more broader discussion of the feedbacks derived here in the context of physical and other biogeochemical feedbacks, as for instance summaried in Ciais et al. 2013.

L500: This is an important caveat that should not be left as a foot note in the conclusion section, as it is a fundamental problem of the approach. I strongly recommend to be more explicit about this in the Methods section, where relevant in Section 4 as well as

specifically in the presentation of Figure 5 and Table 16.

L503: This is a point worth discussing more. Are the feedbacks non-linear and therefore we expect them to be larger/smaller when looking at the difference between present-day and 4xCO2?

L505 and 507: The uncertainties given are the SD of sum of the multi-model mean feedback components, but there are larger uncertainties in the derivation of these feedback that should be discussed and acknowleged.

Data availability: It would be helpful if the authors would list the exact names of the experiments used, including an indication of the ensemble members selected

Please carefully edits and update Table S1

———————————————

---

## Author Comment (AC1) · 22 Jun 2020

Responses to reviews of "Climate-driven chemistry and aerosol feedbacks in CMIP6 Earth system models" by Gillian Thornhill et al.

We would like to thank the two anonymous referees for their useful and supportive comments. Their comments are repeated below reviewer 1 in black, reviewer 2 in blue, with our responses in red.

The paper is an interesting summary of the magnitude of chemistry and aerosol feedbacks in available CMIP6 climate models. The paper is generally well-written, however in its current form the manuscript is somewhat fragmented and some important discussion about is missing. Some aspects of the methodology are described concisely, yet some important details are missing entirely, or are described only briefly. The chemical and aerosol forcing agents are considered independently which helps compartmentalise the results and some of these sections include important insights. However, other sections have not been crafted with the same care.

Thornhill et al. analyse a set of Earth System Model simulations with atmospheric chemistry and aerosol parameterisations to quantify climate feedbacks associated with aerosol and chemistry processes. The methodology allows to attribute the climate feedback to different chemical and aerosol processes and thereby provides in some cases important insights. The paper is highly relevant and fits well to the scope of ACP. The paper is generally well written, but the quality of the individual sections varies considerably.

We thank the reviewers for their positive comments. The comments regarding the fragmentation, missing discussion and individual sections are addressed in responses to specific comments below.

The paper would benefit from merging sections 4 and 5. Currently results from several CMIP6 climate models are somewhat mechanistically portrayed in section 4. Section 5 contains some context for interpreting the differences between models, but uses identical subsection headings and much of the content is more suited to the introduction of a paper on one or more of the forcing agents. The chosen format makes the manuscript unnecessarily disjointed and does not help contextualise the main results. Once sections 4 and 5 are merged, they should be revised to include discussion of the physical processes that cause differences between models. Currently, this is only achieved for one or two of the forcing agents.

I rarely recommend merging results and discussion, but I agree with reviewer #1 that in this case, where a lot of different processes are at play, it would be advisable to merge section 4 and 5 in the sense to have results and discussion for each of the different forcing agents together. The quality of the results presentation and their discussion varies substanially, and the authors should strive to be more explicit in terms of describing and explaining the important differences between models, and where possible provide an appropriate comparison to previous studies. There should then still be an overall discussion section 5/6 in the end where the overall contribution of the non-CO2 chemistry and aerosol feedbacks are discussed in the light of other climate feedbacks (physical, carbon, ...).

We will merge sections 4 and 5 as suggested by both reviewers.

The article has two main themes. Firstly, the differences in aerosol and chemistry forcing efficiency and burden sensitivity are considered. Secondly, the magnitude of feedbacks from forcing agents are

contrasted. It is not clear what the authors intended the main message of the paper to be. The abstract provides very few conclusions about either of these aspects and is overly focussed on methane-specific results. If the paper is intended to focus on the second aspect, then the majority of the feedback summary tables could be moved to the SI without reducing the impact of the paper. However, I think it would be better to retain these tables and include a process-based discussion of the causes of model differences as suggested above.

The focus of the paper is on the quantification of alpha (feedback), which is the product of phi (forcing efficiency) and gamma (sensitivity to climate). Therefore, the phi and gamma terms are equally important. The focus of this paper is not a process-based discussion of model differences. Such discussions could fill whole papers themselves and are to some extent found in the model description papers. Rather the aim of this paper is to demonstrate the contribution of chemistry/aerosol feedback mechanisms to the overall climate feedback in these ESMs. This will be brought out more clearly in the abstract and introduction.

The use of standard deviations to represent uncertainty in a handful of models is not appropriate. It is possible that this is not what the authors have done, but their method is currently unclear. The authors need to clarify their multi-model uncertainty calculations in the text and if they are currently using standard deviations to represent uncertainty in only 3-6 values, need to seek more appropriate ways to communicate this information. Currently multi-model uncertainties are communicated through table captions but should be described fully in the main text.

The methods section should be expanded by a description on how the authors have dealt with uncertainty in this study. What do the reported ± ranges represent for individual estimates, how are errors of the multi-model mean derived from these (error propagation of the IAV?), how is the error range of the total forcing estimate determined, how have varying estimates from emission/burden based methods been dealt with in the total feedback assessment.

We agree the uncertainty methodology needs to be explained fully, including how errors are propagated. This will be added.

The abstract should list the feedbacks assessed here and should be much more explicit about the major findings of this study (I would assume that this would be a summary of Figure 5). It is unclear to me why the methane effects are highlighted here, while this is not mentioned at all in the Conclusion section. In general, the authors should try to clarify the main messages from this paper in abstract, introduction and conclusions.

We realise that the overall aim of the paper was not entirely clear (see also response to reviewer 1). The abstract will be revised to make the findings explicit.

The introduction is somewhat simplistic in that it only lists studies that have attempted to assess non CO2 climate feedbacks. For the general audience and the orientation of the readers it would be helpful to start with a somewhat more detailed description of the major processes and feedbacks considered here and why they matter to the climate system.

We agree this would be a useful addition to the introduction.

The choice of the authors to rely on 4xCO2 experiments to diagnose climate feedback implies that some of the feedbacks considered are less climate change related, but mediated by the effect of CO2 on vegetation productivity and cover. This is an important caveat that should be explained in the Methods section for those processes that do respond to CO2 as well as climate. Also, this needs to be reflected critically in the Conclusions section/Abstract.

This is discussed to some extent in the text, but we agree it could be brought out in the Conclusions and Abstract.

All figures require subfigure labels as per ACP guidelines, to match references in the captions and main text.

These labels will be added.

Line 34: "with warmer temperatures" needs a fuller description. 4Xco2 induced warming

L35: define warmer temperatures

Accepted: "warmer" will be defined.

L36: positive methane feedbacks?

Accepted: This will be reworded

Line 37: VOC needs to be defined.

Accepted: This will be defined.

Line 40: GCMs do these things already. ESMs include the interactions between these systems, by coupling them and hence can expect a greater degree of consistency of information across model components. This needs to be clarified in the text.

Accepted: This will be clarified in the text.

L44: consider adding Arneth et al. 2010, Nat. Geo (Doi: 10.1038/ngeo905) to this list

Accepted: This will be added.

Line 57: Here and in the conclusions, it is important to mention that some of the forcing agents considered make important climate contributions at the regions scale that are neglected when global mean temperatures are used to represent climate change.

Accepted: This will be mentioned.

L72: Briefly explain why you not just use one of the options. I also think that this question deserves more attention in the results section where you for some forcings can compare the magnitude of the alternative estimates more systematically to derive at a joint assessment of the individual feedback factor.

Accepted: The reason for using burdens or emissions will be explained.

Section 2.2: It would be helpful to know which of the feedbacks is calculated which way here. Also, given the need for standardisation here or in the discussion section, there should be a discussion about the assumption of linearity of the radiative forcing response to emissions/burden across a large range of emissions/burden.

Accepted: The discussion of feedbacks will be expanded to include discussion of linearity.

Which ensemble members were selected for this study, or does the study use an ensemble mean?

Only one ensemble member was run for each of these experiments. This will be clarified in the text.

L86: It is unclear whether this is based on simulations presented in Collins et al. 2017, or based on new AerChemMIP experiments, please clarify.

We will clarify that the analysis here is based on simulations from Collins et al. 2017.

Line 102: The scale factor is not well justified. The cited document is a substantial IPCC chapter. Presumably, authors are referring to section 8.2.3.3? Including the page number would help reader. However, the derivation of the scale factor used here is unclear and some explanatory text is required.

L102: Provide an explanation for this scaling factor rather than referencing a full IPCCchapter

Accepted: The scaling factor will be explained.

Line 105: The use of the value 9.25 also needs justification and a description of how it corresponds to values supplied in the referenced document.

L105: For completeness, give value assumed for M_atm as well as the molecular masses of CH4 and air Section 3.1 should reference table 2 but does not. Section 3.2: natural emissions of what?

Accepted: The derivation of the methane lifetime will be explained and the physical constants listed.

Line 110: "four have . . . and three have . . ." is ambiguous. "Three of these four also have. . ." is clearer. Table 2 makes this clear, but is not currently referenced.

Accepted: This will be clarified, and table 2 referenced.

Line 119: Table 3 is currently referenced in a way that suggests it will compare emissions from all natural sources, whereas it actually shows differences between models for dust and BVOCs only. The text needs to be revised. This error is repeated on the first line of section 3.2.2.

L119: this sentence needs to be clarified. There are multiple climate-relevant land based emissions beyond dust and BVOCs. What do the authors want to state here? L134: same as L119

Accepted: The text will be reworded in lines 119 and 134.

Table 3: "PAR" needs to be defined. The phrase "Not dependent on vegetation" is redundant.

Table 3: define LAI, PAR. Given an indication what LAI varies and interactive vegetation imply. The table captions says BVOC, the header VOC, which is correct?

Accepted: LAI and PAR will be defined and the descriptions will be expanded.

Given that Section 4.2.3. discusses wetland emissions, the models used should be described here briefly as well.

Accepted: Wetland will be described as well.

Table 4: There are inconsistencies in the table. For example, sometimes "wind" is used and at other times "wind speed dependent". Descriptions here are too brief. What is the difference between DMS emission and oceanic organic aerosol complexity for NorESM2-LM and UKESM1 for example?

Table 4: what is the difference between wind dependent and wind speed?

Accepted: This table will be reworded for consistency, and descriptions expanded.

L147: Does this sentence imply all models use the same paramterisation?

This will be clarified that the implementation of Price and Rind can vary between models.

Section 4, Line 150: Section 2.1 should be referenced in the first paragraph, so that the normalization of temperatures can be put in the context of $\gamma_i$ as defined in that section.

L151: refer back to Section 2.1 or remove as this is partly redundant. .

Accepted: We agree it would be useful to refer to section 2.1 here.

Line 155: For non-specialist readers an indication of the number of years required to reach equilibrium on average is needed.

L155: For the non-expert reader, explain how long the development of a new equilibrium takes and how large the difference on average would be

Accepted: Yes, we agree this would be useful to indicate.

Table 5: No SD?

We will add standard deviations to this table.

Line 163: Figure S1 does not obviously support this claim. Global mean ERF values should be provided for each model. Also, the authors should explicitly state they are discussing "global mean" effective radiative forcing here.

L163: Figure S1 does not separate shortwave and longwave effects to make this claim.

Accepted: LW and SW will be shown separately. A table of global mean ERFs for each model will be added to the supplement.

Line 165: The strong regional forcing over Africa should be mentioned as the primary cause of positive SW forcing. Some speculation of the process parameterisations that cause this model behavior should be given.

L165: the positive shortwave forcing OF DUST AEROSOLS? Is it possible to provide an explanation for this CNRM response?

Accepted: More explanation will be given here about the absorption values for dust in the different models.

Figure 1 (and subsequent following figures): use stippling or alike to show areas of model dis-/agreement. Also revise figures to ensure the legend is readable without magnifying glasses

Accepted: We will add stippling and increase the size of the legend.

Line 182: Refer to table 6 again. Also, some speculation on the physical processes causing the increased lifetime should be given. This is a good example of the need for additional discussion and how merging, then adapting content from section 5 will improve the interpretation of results. Line 189: It is not clear what the 2nd use of "for instance" here is referring to. This sentence needs to be rewritten to improve clarity.

Accepted: Reference to table 6 will be added along with a comment on the physical processes. The 2nd "for instance" was a mistake and the sentence will be rewritten.

Table 6: The reason for missing values in this and other tables needs to be explained more clearly within the text.

Table 6 (and similar subsequent tables): Why are certain cells blank?

Accepted: Not all models provided all the diagnostics. This will be explained.

Line 200: These forcing values are far larger than for dust. Are the forcing-emission-feedback relationships expected to be linear? If not, there will be discrepancies in the gamma terms across emission types, even if normalised. This assumption on linearity and its implications need to be discussed here and/or in section 2.2.

A comment on the non-linearity will be added. For doubled emissions, errors introduced through assuming linearity are likely to be small compared to process uncertainty. Many studies use 5x or 10x emissions.

Line 201: Why 20x? Is this caused by the choice of size bins? This warrants some discussion. Why is the AOD of a similar magnitude? What model processes have been adjusted/tuned to make the AOD similar? The reasons models have similar values for very different reasons need to be better understood. This is important for understanding the causes of model diversity in climate projections.

L201: Why does this discrepancy occur, and how can the AOD be still similar? This paragraph should also have a discussion on why MIROC6 deviates in terms of the ERF response

This is indeed due to CNRM having a bin for larger particles (up to 20 microns) which add to the mass, but not to the AOD. This will be clarified.

Line 205 - 209: All positive except MIROC needs to be explained/considered. What regions show a decrease in emissions that causes the global mean response to be negative? Maps for each model in the SI are needed.

Accepted: Maps for each model will be provided in the supplement, and the regionality of the MIROC decrease (decrease in N. hemisphere especially N. Atlantic, just outweighs the increases in the Southern Ocean) will be described.

Line 220: It should be explained here that all models could have run the 2xdms experiment. Interactive ocean biogeochemistry is not a prerequisite, since emissions could have been scaled within the flux parameterisation as with the 2xdust experiment.

Accepted: This will be clarified.

Line 222: Fig 3 does not show the forcing values for each model as implied. Table 8 should be referenced to here.

L222: Figure 3 does not show this.

Accepted: The text will be clarified that fig 3 shows the multi-model mean. Table 8 will be referenced.

Line 222-224: Maps of sulphur concentrations and changes in concentrations need to be included as a figure in the SI for each model, so the reader has a clear understanding of the magnitude of regional compensation across models.

Accepted: These maps will be added.

Line 225: GFDL-ESM4 values only contributes to the multi-model sensitivity to emissions/concentrations, but not to the multi-model radiative efficiency. The assumption made here is that all models have similar radiative efficiencies. This is an important assumption, given the diversity of model responses highlighted up to this point in the manuscript. Is it appropriate to assume GFDL-ESM4 has the same radiative efficiency as the two models used in the sensitivity calculation? Some justification is required if the authors want to maintain this approach. An alternative approach would be to only use the 2 models with sufficient information to calculate both the multi-model sensitivity and multi-model radiative efficiency. This subjective choice to include partial information from one model needs to be justified more clearly and the implications of extrapolating the multi-model radiative efficiency to other models needs to be considered and openly discussed.

Accepted: This is a good point and both methods will be investigated.

Line 229: The magnitude of the increase should be quantified in the text.

Accepted: This will be added to the text.

Table 8: Please check values, at least the alpha emission multi model mean cannot be correct.

These values will be recalculated.

Line 249: Here and elsewhere in the text, the word "significant" is used without mention of associated statistical tests. The values should be state with "significant" removed, or the methodology more accurately described.

Accepted: This will be more accurately described.

Line 253: Incorrect label. Figure S3 only shows the multi-model mean. Given the diversity in aerosol forcing from this source, maps of CDNC should be provided for each model. Also, interpretation of the differences between models needs to be included here.

L253: Figure S4 does not exist.

Apologies for the mistake in the figure reference. CDNC will be shown for each model.

Line 253-257: Examples of regions where these behaviors are likely, with an explanation of why is needed.

This explanation will be expanded.

Figure 4: Fig S3 could be a subfigure of Fig 4.

Agreed the multi-model CDNC could fit in figure 4.

Table 9: Uncertainty values are missing for UKESM1 and multi-model mean values are missing for Scaled Mass

These will be fixed.

Section 4.2 general: I think it would be easier to follow if the indirect effects of NOx and BVOC on methane were discussed jointly and possible even in one table, as they rely on the same methodology and type of experiments.

Accepted: These will be put into the same table.

Line 278: Is there an hypothesis the authors could provide to explain the causes of model diversity in BVOC partitioning into ozone and aerosol forcing? This sort of discussion is essential to develop a better understanding of the importance model differences and will affect interpretation of climate feedbacks across models.

This isn't a partitioning as such as the production of ozone and SOA are through very different mechanisms. The ozone responses are similar (in DU per Tg/yr VOC), but the SOA varies more. Some discussion will be added on why the models might differ.

L279: BVOC-related aerosols, or aerosols in general?

This will be clarified that this relates to the aerosols from BVOC changes.

L281: refer back to Section 2.2.

Accepted: A reference to section 2.2 will be added.

Table 10: There is no explanation of why 14% is used. This should be in the methods section, not hidden in a caption.

A justification of the radiative efficiency uncertainty will be provided in the main text.

L297: Methane Burden/Emissions? does not change

Methane concentration does not change. This will be clarified.

Line 300-302: This sentence needs to be rewritten to improve readability.

This will be rewritten.

Line 302: It is not clear from the text as written, how BVOC burden sensitivities are used in the methane sensitivity calculation.

This will be clarified – it is sensitivity of methane lifetime to BVOC emission in the 2xVOC experiment.

L304: The 0.015 Wm-2 %-1 are not described in Section 2.2. but should be

Accepted: These will be described in section 2.2

Section 4.2.4: The title of this section is misleading. Several non-emission drivers are considered, not just these two.

This will be clarified to be "Meteorological Drivers"

Line 265: "1" missing from UKESM1.

This will be corrected.

Section 4.2.3 I find this section troublesome given the lack of explanation of the simulated methane emissions, particular because this presentation confounds the direct effects of CO2 on methane emissions (via CO2 fertilisation of wetlands) with the direct effects of temperature on methane-emissions, but exclusively attributes this to temperature. The result of which is an inflated methane-emission climate feedback compared to Ciais et al. 2013. I wonder whether there are simulations with interactive methane but no biogeochemical coupling to CO2 available from the C4MIP project that would allow to tackle this separation? As a minimum, this confounding effect needs to be explained and discussed.

Unfortunately there are no radiation-only 4xCO2 simulations. The effects will be explained and discussed.

Table 14: What is the justification to assume at 14% uncertainty on methane radiative efficiency? Section 4.2.4 should be labelled atmospheric temperature and water vapour?

This will be justified (see also review 1 comment on table 10).

L356: the residual is then ASSUMED TO BE the direct effect. This statement could be backed up by a brief explanation that BVOC and NOx are the only agents affecting ozone and methane lifetimes next

to climate in these models. Otherwise, it should be explained why other factors may be small and negligible.

Accepted: A brief explanation will be given.

L367: Consistently use CESM-WACCM

Accepted: We will check for naming consistency.

Section 4.3: This section needs some comment about the importance of climate forcing agents that have climatic importance at the regional scale, to prevent the results of this manuscript being interpreted incorrectly.

The focus of the paper is on the global radiative feedback per K, but we agree that it would be useful to clarify that there could be regional responses to a global forcing.

Section 4.3: Figure 5 is not referenced. The text needs to be explicit that the feedbacks are the multi-model mean, and that not all feedbacks could be calculated for all processes considered. A discussion that I have been missing here is whether these terms are really additive and linear as assumed. It is possible that there is a compensation of feedbacks between models, so I wonder whether it would be possible / interesting to compare the sum of feedbacks across processes for those models that have calculated similar feedbacks

It is not obvious that there would be significant lack of additivity given these are small changes in composition, however we will add a discussion.

Line 380: The authors need to specify that these are multi-model feedbacks, here and in the table caption. Figure 5 needs to be referenced. In addition, the cancellation between models with opposite signs again needs to be mentioned within this section, as does the fact that a different number of models were used to calculate the multimodel means because of data availability.

These are all good points and will be implemented.

Figure 5: use consistent labelling of models. use consistent labelling of forcing factors (e.g. total non-CH4, wetland CH4 etc.) Use a clearer abbreviation for lightning NOx than lNOx. The figure caption should also explain, how and why feedbacks from table 16 were aggregated in the figure.

Accepted: The labelling and caption will be improved.

Line 402: Can the feedbacks be interpreted in the context of the magnitude of forcing from these forcing agents over some specified period? Uncertainty in these magnitudes should be included in the discussion with appropriate references.

The forcing responses are maintained continuously rather than being for a specific period.

Line 423: There is no use citing these values if not directly comparable. This text should be removed to avoid confusion. Further discussion of the causes of model differences is required here.

Section 5.2 is not helpful is no guidance is given as to the origin of the large range in the estimates and the plausiblity of the different model projections. The comparison to the literature numbers is insufficient in that the numbers aren't directly comparable. This section needs substanial revision.

This will be rewritten to discuss what comparisons are available.

Line 433: Please clarify the difference between primary production and DMS production in the text.

Accepted: This will be clarified.

Section 5.6 response to my previous comment, but then implies that this shouldn't really be listed here as a climate feedback, but a biogeochemical carbon-methane feedback.

We describe this here as an "adjustment", but we agree we need to make it more explicit in the methods and conclusions that it is not necessarily a feedback.

Section 6: I would have liked to see a somewhat more broader discussion of the feedbacks derived here in the context of physical and other biogeochemical feedbacks, as for instance summaried in Ciais et al. 2013.

Accepted: We will expand section 6 to include more context.

L500: This is an important caveat that should not be left as a foot note in the conclusion section, as it is a fundamental problem of the approach. I strongly recommend to be more explicit about this in the Methods section, where relevant in Section 4 as well as specifically in the presentation of Figure 5 and Table 16.

Accepted: As with the section 5.6 comment a discussion of "adjustments" will be made more explicit in the Methods.

L503: This is a point worth discussing more. Are the feedbacks non-linear and therefore we expect them to be larger/smaller when looking at the difference between present-day and 4xCO2?

The choice of base state is likely to be important for the forcing efficiencies. We might expect aerosol forcing to be less efficient and ozone production more efficient in the present day. A discussion of this will be added.

Line 505: This value needs context to aid interpretation. e.g. What is this as a proportion of the GHG forcing required to increase temperatures by 1 degree?

Good point: We will add a comparison to the total climate feedback ~ -1.25 W/m2/K and climate senstivity.

L505 and 507: The uncertainties given are the SD of sum of the multi-model mean feedback components, but there are larger uncertainties in the derivation of these feedback that should be discussed and acknowleged.

Line 507-508: The uncertainties in these values are substantial and need to be included in this discussion and interpretation of results.

Accepted: The discussion of the uncertainties will be expanded.

SI: S1, some descriptions are missing entirely and need to be included.

These will be added.

SI: All figures require subfigure labels.

These will be added.

Data availability: It would be helpful if the authors would list the exact names of the experiments used, including an indication of the ensemble members selected Please carefully edits and update Table S1

Table S1 will be expanded to include this information.

---

## Author Response (AR1)

Responses to reviews of "Climate-driven chemistry and aerosol feedbacks in CMIP6 Earth system models" by Gillian Thornhill et al.

We would like to thank the two anonymous referees for their useful and supportive comments. Their comments are repeated below reviewer 1 in black, reviewer 2 in blue, with our responses in red.

5

The paper is an interesting summary of the magnitude of chemistry and aerosol feedbacks in available CMIP6 climate models. The paper is generally well-written, however in its current form the manuscript is somewhat fragmented and some important discussion about is missing. Some aspects of the methodology are described concisely, yet some important details are missing entirely, or are described only briefly. The chemical and aerosol forcing agents are considered independently which helps

10 compartmentalise the results and some of these sections include important insights. However, other sections have not been crafted with the same care.

Thornhill et al. analyse a set of Earth System Model simulations with atmospheric chemistry and aerosol parameterisations to quantify climate feedbacks associated with aerosol and chemistry processes. The methodology allows to attribute the climate feedback to different chemical and aerosol processes and thereby provides in some cases important insights. The paper is

15 highly relevant and fits well to the scope of ACP. The paper is generally well written, but the quality of the individual sections varies considerably.

We thank the reviewers for their positive comments. The comments regarding the fragmentation, missing discussion and individual sections are addressed in responses to specific comments below.

- 20 The paper would benefit from merging sections 4 and 5. Currently results from several CMIP6 climate models are somewhat mechanistically portrayed in section 4. Section 5 contains some context for interpreting the differences between models, but uses identical subsection headings and much of the content is more suited to the introduction of a paper on one or more of the forcing agents. The chosen format makes the manuscript unnecessarily disjointed and does not help contextualise the main results. Once sections 4 and 5 are merged, they should be revised to include discussion of the physical processes that cause
- 25 differences between models. Currently, this is only achieved for one or two of the forcing agents. I rarely recommend merging results and discussion, but I agree with reviewer #1 that in this case, where a lot of different processes are at play, it would be advisable to merge section 4 and 5 in the sense to have results and discussion for each of the different forcing agents together. The quality of the results presentation and their discussion varies substanially, and the authors should strive to be more explicit in terms of describing and explaining the important differences between models, and where
- 30 possible provide an appropriate comparison to previous studies. There should then still be an overall discussion section 5/6 in the end where the overall contribution of the non-CO2 chemistry and aerosol feedbacks are discussed in the light of other climate feedbacks (physical, carbon, ...).

We have merged sections 4 and 5 as suggested by both reviewers.

- 35 The article has two main themes. Firstly, the differences in aerosol and chemistry forcing efficiency and burden sensitivity are considered. Secondly, the magnitude of feedbacks from forcing agents are contrasted. It is not clear what the authors intended the main message of the paper to be. The abstract provides very few conclusions about either of these aspects and is overly focussed on methane-specific results. If the paper is intended to focus on the second aspect, then the majority of the feedback summary tables could be moved to the SI without reducing the impact of the paper. However, I think it would be better to retain these tables and include a process-based discussion of the causes of model differences as suggested above.
- We have rewritten the abstract to more closely reflect the structure and findings of the paper. The focus of the paper is on the quantification of alpha (feedback), which is the product of phi (forcing efficiency) and gamma (sensitivity to climate). Therefore, the phi and gamma terms are equally important. The focus of this paper is not a process-based discussion of model differences. Such discussions could fill whole papers themselves and are to some extent found in the model description papers.
- 45 Rather the aim of this paper is to demonstrate the contribution of chemistry/aerosol feedback mechanisms to the overall climate feedback in these ESMs. This has brought out more clearly in the abstract and introduction.

The use of standard deviations to represent uncertainty in a handful of models is not appropriate. It is possible that this is not what the authors have done, but their method is currently unclear. The authors need to clarify their multi-model uncertainty

50 calculations in the text and if they are currently using standard deviations to represent uncertainty in only 3-6 values, need to seek more appropriate ways to communicate this information. Currently multi-model uncertainties are communicated through table captions but should be described fully in the main text.

The methods section should be expanded by a description on how the authors have dealt with uncertainty in this study. What do the reported  $\pm$  ranges represent for individual estimates, how are errors of the multi-model mean derived from these (error

55 propagation of the IAV?), how is the error range of the total forcing estimate determined, how have varying estimates from emission/burden based methods been dealt with in the total feedback assessment. We have explained more fully where we have used interannual variability and where inter-model variability. The abstract should list the feedbacks assessed here and should be much more explicit about the major findings of this study

(I would assume that this would be a summary of Figure 5). It is unclear to me why the methane effects are highlighted here,

60 while this is not mentioned at all in the Conclusion section. In general, the authors should try to clarify the main messages from this paper in abstract, introduction and conclusions.

We realise that the overall aim of the paper was not entirely clear (see also response to reviewer 1). We have substantially revised the abstract to convey the main messages.

The introduction is somewhat simplistic in that it only lists studies that have attempted to assess non CO2 climate feedbacks.

65 For the general audience and the orientation of the readers it would be helpful to start with a somewhat more detailed description of the major processes and feedbacks considered here and why they matter to the climate system.

We agree this is a useful addition to the introduction. We have added a few sentences of text and references to Sherwood et al. 2020 and Friedlingstein 2015.

The choice of the authors to rely on 4xCO2 experiments to diagnose climate feedback implies that some of the feedbacks

- 70 considered are less climate change related, but mediated by the effect of CO2 on vegetation productivity and cover. This is an important caveat that should be explained in the Methods section for those processes that do respond to CO2 as well as climate. Also, this needs to be reflected critically in the Conclusions section/Abstract. This was discussed to some extent in the main text. This has been brought out in the Conclusions.
- 75 All figures require subfigure labels as per ACP guidelines, to match references in the captions and main text. These labels have been added.

Line 34: "with warmer temperatures" needs a fuller description. 4Xco2 induced warming L35: define warmer temperatures

80 Accepted: "warmer" has been rephrased as "warmer surface temperatures following a quadrupling of CO2 concentrations"

L36: positive methane feedbacks? Accepted: This paragraph has been completely reworded.

Line 37: VOC needs to be defined.Accepted: BVOC has been defined.

Line 40: GCMs do these things already. ESMs include the interactions between these systems, by coupling them and hence can expect a greater degree of consistency of information across model components. This needs to be clarified in the text.

**90** Accepted: This has been clarified in the text. "Earth system models extend the complexity of physical climate models by coupling land and ocean biospheres, atmospheric chemistry and aerosols to the physical climate"

L44: consider adding Arneth et al. 2010, Nat. Geo (Doi: 10.1038/ngeo905) to this list Accepted: This has been added.

95

Line 57: Here and in the conclusions, it is important to mention that some of the forcing agents considered make important climate contributions at the regions scale that are neglected when global mean temperatures are used to represent climate change.

Accepted: We have added sentences mentioning this to the introduction and conclusions.

3

100 L72: Briefly explain why you not just use one of the options. I also think that this question deserves more attention in the results section where you for some forcings can compare the magnitude of the alternative estimates more systematically to derive at a joint assessment of the individual feedback factor.

Accepted: We have added text in section 2.2 to explain the reasons for using burdens, AODs or emissions for different species.

- 105 Section 2.2: It would be helpful to know which of the feedbacks is calculated which way here. Also, given the need for standardisation here or in the discussion section, there should be a discussion about the assumption of linearity of the radiative forcing response to emissions/burden across a large range of emissions/burden. Accepted: The discussion of feedbacks has been expanded in section 2.2 to include discussion of linearity.
- 110 Which ensemble members were selected for this study, or does the study use an ensemble mean? Only one ensemble member was run for each of these experiments. This has been clarified in the text.

L86: It is unclear whether this is based on simulations presented in Collins et al. 2017, or based on new AerChemMIP experiments, please clarify.

115 We have clarified that the analysis here is based on simulations from Collins et al. 2017.

Line 102: The scale factor is not well justified. The cited document is a substantial IPCC chapter. Presumably, authors are referring to section 8.2.3.3? Including the page number would help reader. However, the derivation of the scale factor used here is unclear and some explanatory text is required.

120 L102: Provide an explanation for this scaling factor rather than referencing a full IPCCchapter Accepted: More detail on this scaling factor is added to this section. The precise section (8.SM.11.3.2) is provided.

Line 105: The use of the value 9.25 also needs justification and a description of how it corresponds to values supplied in the referenced document.

125 L105: For completeness, give value assumed for M\_atm as well as the molecular masses of CH4 and air Accepted: The derivation of the methane lifetime has been explained and the physical constants listed with a reference to Prather et al. 2012

Section 3.1 should reference table 2 but does not.

A reference to table 2 has been added.

130 Section 3.2: natural emissions of what?This has been clarified to be "of aerosols and ozone precursors".

Line 110: "four have . . . and three have . . ." is ambiguous. "Three of these four also have. . ." is clearer. Table 2 makes this clear, but is not currently referenced.

Accepted: This has been clarified with a reference to table 2.

135 Line 119: Table 3 is currently referenced in a way that suggests it will compare emissions from all natural sources, whereas it actually shows differences between models for dust and BVOCs only. The text needs to be revised. This error is repeated on the first line of section 3.2.2.

L119: this sentence needs to be clarified. There are multiple climate-relevant land based emissions beyond dust and BVOCs. What do the authors want to state here? L134: same as L119

140 Accepted: This has been changed to make it clear that these are the emissions analysed in this study, rather than making any more general claim.

Table 3: "PAR" needs to be defined. The phrase "Not dependent on vegetation" is redundant.

Table 3: define LAI, PAR. Given an indication what LAI varies and interactive vegetation imply. The table captions says

145 BVOC, the header VOC, which is correct?

Accepted: LAI and PAR have defined and the descriptions expanded. The header has been changed to BVOC

Given that Section 4.2.3. discusses wetland emissions, the models used should be described here briefly as well. Accepted: Wetland models have been described as well.

150

Table 4: There are inconsistencies in the table. For example, sometimes "wind" is used and at other times "wind speed dependent". Descriptions here are too brief. What is the difference between DMS emission and oceanic organic aerosol complexity for NorESM2-LM and UKESM1 for example?

155 Table 4: what is the difference between wind dependent and wind speed?Accepted: This table has been reworded for consistency, and descriptions expanded.

L147: Does this sentence imply all models use the same paramterisation?

This has been clarified that the implementation of Price and Rind can vary between models.

160

Section 4, Line 150: Section 2.1 should be referenced in the first paragraph, so that the normalization of temperatures can be put in the context of  $\gamma$ i as defined in that section.

L151: refer back to Section 2.1 or remove as this is partly redundant. .

Accepted: We agree reference to  $\gamma_i$  and section 2.1 has been added.

165

Line 155: For non-specialist readers an indication of the number of years required to reach equilibrium on average is needed. L155: For the non-expert reader, explain how long the development of a new equilibrium takes and how large the difference on average would be

Accepted: Yes, we added a comment on this taking many centuries.

**170**

175

180

Table 5: No SD?

We have added standard deviations to this table.

Line 163: Figure S1 does not obviously support this claim. Global mean ERF values should be provided for each model. Also, the authors should explicitly state they are discussing "global mean" effective radiative forcing here.

L163: Figure S1 does not separate shortwave and longwave effects to make this claim. Accepted: We have added separate maps of LW and SW in the supplement.

Line 165: The strong regional forcing over Africa should be mentioned as the primary cause of positive SW forcing. Some speculation of the process parameterisations that cause this model behavior should be given.

L165: the positive shortwave forcing OF DUST AEROSOLS? Is it possible to provide an explanation for this CNRM response? Accepted: More description has been added on where models agree or disagree in the LW and SW forcing.

Figure 1 (and subsequent following figures): use stippling or alike to show areas of model dis-/agreement. Also revise figures

185 to ensure the legend is readable without magnifying glasses Accepted: We have added stippling and increased the size of the legend.

Line 182: Refer to table 6 again. Also, some speculation on the physical processes causing the increased lifetime should be given. This is a good example of the need for additional discussion and how merging, then adapting content from section 5

190 will improve the interpretation of results. Line 189: It is not clear what the 2nd use of "for instance" here is referring to. This sentence needs to be rewritten to improve clarity.

Accepted: Reference to table 6 has been added along with a comment on the physical processes. The  $2^{nd}$  "for instance" was a mistake and the sentence has been rewritten.

Table 6: The reason for missing values in this and other tables needs to be explained more clearly within the text.Table 6 (and similar subsequent tables): Why are certain cells blank?Accepted: Not all models provided all the diagnostics. These have been filled with N/A

Line 200: These forcing values are far larger than for dust. Are the forcing-emission-feedback relationships expected to be

200 linear? If not, there will be discrepancies in the gamma terms across emission types, even if normalised. This assumption on linearity and its implications need to be discussed here and/or in section 2.2.

A comment on the non-linearity has been added to the introduction. For doubled emissions, errors introduced through assuming linearity are likely to be small compared to process uncertainty. Many studies use 5x or 10x emissions.

205 Line 201: Why 20x? Is this caused by the choice of size bins? This warrants some discussion. Why is the AOD of a similar magnitude? What model processes have been adjusted/tuned to make the AOD similar? The reasons models have similar values for very different reasons need to be better understood. This is important for understanding the causes of model diversity in climate projections.

L201: Why does this discrepancy occur, and how can the AOD be still similar? This paragraph should also have a discussion on why MIROC6 deviates in terms of the ERF response

210 on why MIROC6 deviates in terms of the ERF response This is indeed due to CNRM having a bin for larger particles (up to 20 microns) which add to the mass, but not to the AOD. This has been clarified in the text.

Line 205 - 209: All positive except MIROC needs to be explained/considered. What regions show a decrease in emissions that causes the global mean response to be negative? Maps for each model in the SI are needed.

Accepted: Maps for each model have been provided in the supplement. "The global mean change in emissions is positive in all models except MIROC6 and GISS-E2-1 (where the lower latitude decreases outweigh the high latitude increases). " Line 220: It should be explained here that all models could have run the 2xdms experiment. Interactive ocean biogeochemistry is not a prerequisite, since emissions could have been scaled within the flux parameterisation as with the 2xdust experiment.

220 Accepted: This has been clarified.

Line 222: Fig 3 does not show the forcing values for each model as implied. Table 8 should be referenced to here.

L222: Figure 3 does not show this.

Accepted: The text will has been clarified that fig 3 shows the multi-model mean, individual maps have been added to the supplement. Table 8 has been referenced.

Line 222-224: Maps of sulphur concentrations and changes in concentrations need to be included as a figure in the SI for each model, so the reader has a clear understanding of the magnitude of regional compensation across models. Accepted: Individual maps of emissions and ERF have been added to the supplement.

230

225

Line 225: GFDL-ESM4 values only contributes to the multi-model sensitivity to emissions/concentrations, but not to the multimodel radiative efficiency. The assumption made here is that all models have similar radiative efficiencies. This is an important assumption, given the diversity of model responses highlighted up to this point in the manuscript. Is it appropriate to assume GFDL-ESM4 has the same radiative efficiency as the two models used in the sensitivity calculation? Some justification is

235 required if the authors want to maintain this approach. An alternative approach would be to only use the 2 models with sufficient information to calculate both the multi-model sensitivity and multi-model radiative efficiency. This subjective choice to include partial information from one model needs to be justified more clearly and the implications of extrapolating the multi-model radiative efficiency to other models needs to be considered and openly discussed.

Accepted: This is a good point, for consistency we decided to use only the 2 models with sufficient information to calculate 240 both the multi-model sensitivity and multi-model radiative efficiency

Line 229: The magnitude of the increase should be quantified in the text. Accepted: The increase in lifetime has been added to the text.

245Table 8: Please check values, at least the alpha emission multi model mean cannot be correct.All values have been recalculated.

Line 249: Here and elsewhere in the text, the word "significant" is used without mention of associated statistical tests. The values should be state with "significant" removed, or the methodology more accurately described.

250 Accepted: Significant has been replaced with more appropriate wording unless it specifically refers to a statistical test.

Line 253: Incorrect label. Figure S3 only shows the multi-model mean. Given the diversity in aerosol forcing from this source, maps of CDNC should be provided for each model. Also, interpretation of the differences between models needs to be included here.

255 L253: Figure S4 does not exist.

This figure came from only one model. We do not have sufficient data from all the models to make CDNC claims so this sentence and figure has been removed.

Line 253-257: Examples of regions where these behaviors are likely, with an explanation of why is needed.

260 We do not have sufficient data from all the models to explore these points, so these sentences have been removed.

Figure 4: Fig S3 could be a subfigure of Fig 4. This figure has been removed. 265 Table 9: Uncertainty values are missing for UKESM1 and multi-model mean values are missing for Scaled Mass This table has been revised.

Section 4.2 general: I think it would be easier to follow if the indirect effects of NOx and BVOC on methane were discussed jointly and possible even in one table, as they rely on the same methodology and type of experiments.

270 Accepted: These have put into the same table (table 12).

Line 278: Is there an hypothesis the authors could provide to explain the causes of model diversity in BVOC partitioning into ozone and aerosol forcing? This sort of discussion is essential to develop a better understanding of the importance model differences and will affect interpretation of climate feedbacks across models.

275 This isn't a partitioning as such as the production of ozone and SOA are through very different mechanisms. Discussion has to be added to the effect that the ozone responses are similar, but the SOA varies more.

L279: BVOC-related aerosols, or aerosols in general?

This has been clarified that this relates to the aerosols from BVOC changes.

280

L281: refer back to Section 2.2.

Accepted: A reference to section 2.2 has been added.

Table 10: There is no explanation of why 14% is used. This should be in the methods section, not hidden in a caption.285The reference to Etminan has been added to explain the 14% uncertainty.

L297: Methane Burden/Emissions? does not change This will be clarified to say that the methane concentration does not change.

290 Line 300-302: This sentence needs to be rewritten to improve readability. This whole discussion has been rewritten.

Line 302: It is not clear from the text as written, how BVOC burden sensitivities are used in the methane sensitivity calculation. This has been clarified in the text – it is sensitivity of methane lifetime to BVOC emission in the 2xVOC experiment.

295

L304: The 0.015 Wm-2 %-1 are not described in Section 2.2. but should be

Accepted: The conversion of lifetime change to ERF change has been added to section 2.2.

Section 4.2.4: The title of this section is misleading. Several non-emission drivers are considered, not just these two. This has been changed to be "Meteorological Drivers"

Line 265: "1" missing from UKESM1. This has been corrected

300

- 305 Section 4.2.3 I find this section troublesome given the lack of explanation of the simulated methane emissions, particular because this presentation confounds the direct effects of CO2 on methane emissions (via CO2 fertilisation of wetlands) with the direct effects of temperature on methane-emissions, but exclusively attributes this to temperature. The result of which is an inflated methane-emission climate feedback compared to Ciais et al. 2013. I wonder whether there are simulations with interactive methane but no biogeochemical coupling to CO2 available from the C4MIP project that would allow to tackle this
- 310 separation? As a minimum, this confounding effect needs to be explained and discussed. Unfortunately there are no radiation-only 4xCO2 simulations. We have caveated the wetland results, particularly as we have only two models.

Table 14: What is the justification to assume at 14% uncertainty on methane radiative efficiency? Section 4.2.4 should be labelled atmospheric temperature and water vapour?

We have added a reference to section 2.2. which now describes where this uncertainty comes from. The section has ben renamed Meteorological drivers.

L356: the residual is then ASSUMED TO BE the direct effect. This statement could be backed up by a brief explanation that
BVOC and NOx are the only agents affecting ozone and methane lifetimes next to climate in these models. Otherwise, it should be explained why other factors may be small and negligible.
Accepted: An explanation of this has been added to the text.

L367: Consistently use CESM-WACCM

325 Accepted: We have checked for naming consistency.

Section 4.3: This section needs some comment about the importance of climate forcing agents that have climatic importance at the regional scale, to prevent the results of this manuscript being interpreted incorrectly.

The focus of the paper is on the global radiative feedback per K, but we have added text "This analysis (and climate sensitivity

330 in general) is focussed on the global mean, but it should be noted that the cooling effects of increased aerosols will be heterogenous and some regions will experience less warming than a global climate sensitivity might suggest"

Section 4.3: Figure 5 is not referenced. The text needs to be explicit that the feedbacks are the multi-model mean, and that not all feedbacks could be calculated for all processes considered. A discussion that I have been missing here is whether these

335 terms are really additive and linear as assumed. It is possible that there is a compensation of feedbacks between models, so I wonder whether it would be possible / interesting to compare the sum of feedbacks across processes for those models that have calculated similar feedbacks

Figure 5 has been referenced. It is not obvious that there would be significant lack of additivity given these are small changes in composition, however we have added a discussion. "The totals assume that feedbacks are additive, which is the basis of the framework in section 2.1."

Line 380: The authors need to specify that these are multi-model feedbacks, here and in the table caption. Figure 5 needs to be referenced. In addition, the cancellation between models with opposite signs again needs to be mentioned within this section, as does the fact that a different number of models were used to calculate the multimodel means because of data availability.

345 These are all good points and have been implemented.

Figure 5: use consistent labelling of models. use consistent labelling of forcing factors (e.g. total non-CH4, wetland CH4 etc.) Use a clearer abbreviation for lightning NOx than lNOx. The figure caption should also explain, how and why feedbacks from table 16 were aggregated in the figure.

350 Accepted: The labelling has been changed, and the caption now describes how feedbacks are aggregated.

Line 402: Can the feedbacks be interpreted in the context of the magnitude of forcing from these forcing agents over some specified period? Uncertainty in these magnitudes should be included in the discussion with appropriate references. The forcing responses are maintained continuously rather than being for a specific period.

355

340

Line 423: There is no use citing these values if not directly comparable. This text should be removed to avoid confusion. Further discussion of the causes of model differences is required here.

Section 5.2 is not helpful is no guidance is given as to the origin of the large range in the estimates and the plausibility of the different model projections. The comparison to the literature numbers is insufficient in that the numbers aren't directly

360 comparable. This section needs substanial revision.

This discussion has been removed as the numbers aren't directly comparable.

Line 433: Please clarify the difference between primary production and DMS production in the text.

Accepted: This is now referred to as biological activity

365

Section 5.6 response to my previous comment, but then implies that this shouldn't really be listed here as a climate feedback, but a biogeochemical carbon-methane feedback.

We describe this here as an "adjustment", but we have made it more explicit in the methods and conclusions that it is not necessarily a feedback.

370

Section 6: I would have liked to see a somewhat more broader discussion of the feedbacks derived here in the context of physical and other biogeochemical feedbacks, as for instance summaried in Ciais et al. 2013. Accepted: We have compared to both Ciais 2013 and Sherwood 2020 to provide a comparison with other feedbacks.

- 375 L500: This is an important caveat that should not be left as a foot note in the conclusion section, as it is a fundamental problem of the approach. I strongly recommend to be more explicit about this in the Methods section, where relevant in Section 4 as well as specifically in the presentation of Figure 5 and Table 16.
  Accepted: As with the section 5.6 comment a discussion of "adjustments" has been made more explicit in the Methods.
- 380 L503: This is a point worth discussing more. Are the feedbacks non-linear and therefore we expect them to be larger/smaller when looking at the difference between present-day and 4xCO2? The choice of base state is likely to be important for the forcing efficiencies. We might expect aerosol forcing to be less efficient and ozone production more efficient in the present day. This is now mentioned in the text.
- 385 Line 505: This value needs context to aid interpretation. e.g. What is this as a proportion of the GHG forcing required to increase temperatures by 1 degree?

Accepted: We have compared to both Ciais 2013 and Sherwood 2020 to provide a comparison with other feedbacks.

L505 and 507: The uncertainties given are the SD of sum of the multi-model mean feedback components, but there are larger uncertainties in the derivation of these feedback that should be discussed and acknowleged.

Line 507-508: The uncertainties in these values are substantial and need to be included in this discussion and interpretation of results.

Accepted: A discussion of possible systematic uncertainties has been added to section 4.3

395 SI: S1, some descriptions are missing entirely and need to be included.These descriptions have been added.

SI: All figures require subfigure labels.

We do not refer to specific subfigures individually here.

**400**

Data availability: It would be helpful if the authors would list the exact names of the experiments used, including an indication of the ensemble members selected Please carefully edits and update Table S1 The exact names are as listed in table 1. Table S2 has been added to include this information.

**405 Climate-driven chemistry and aerosol feedbacks in CMIP6 Earth system models**

Gillian Thornhill1, William Collins1, Dirk Olivié2, Ragnhild B. Skeie3, Alex Archibald4,5, Susanne Bauer6, Ramiro Checa-Garcia7, Stephanie Fiedler8, Gerd Folberth9, Ada Gjermundsen2, Larry Horowitz10, Jean-Francois Lamarque11, Martine Michou12, Jane Mulcahy9, Pierre Nabat12, Vaishali Naik10, Fiona M. O'Connor9, Fabien Paulot10, Michael Schulz2, Catherine E. Scott13, Roland Seferian12, Chris Smith13,

410 O'Connor9, Fabien Paulot10, Michael Schulz2, Catherine E. Scott13, Roland Seferian12, Chris Smit Toshihiko Takemura14, Simone Tilmes11, Kostas Tsigaridis6,15, James Weber4,

1Department of Meteorology, University of Reading, Reading, RG6 6BB, UK 2Norwegian Meteorological Institute, Oslo, Norway 415 3CICERO – Centre for International Climate and Environmental Research Oslo, Oslo, Norway 4Department of Chemistry, University of Cambridge, Cambridge, CB2 1EW, UK 5National Centre for Atmospheric Science, UK 6NASA Goddard Institute for Space Studies, 2880 Broadway 1 IPSL/LSCE CEA-CNRS-UVSO-UPSaclay UMR Gif sur Yvette, FRANCE 420 8Max-Planck-Institute for Meteorology, Hamburg, 20146, Germany 9Met 
[revised manuscript text omitted]
-seven models have interactive aerosol schemes, four-five have interactive stratospheric chemistry and-four of whichthree also have interactive tropospheric chemistry (table 2). The level of sophistication of the chemistry can affect the modelled responses to the emissions of reactive gases. For instance, in models without interactive tropospheric chemistry changes in biogenic volatile organic compound emissions (BVOCs) affect only organic aerosols, whereas in models with interactive tropospheric chemistry they also affect ozone, methane lifetime, and potentially the oxidation of other aerosol precursors. For each model one ensemble member was run for each experiment.

|             | Tropospheric chemistry | Stratospheric chemistry | Reference                                        |
|-------------|------------------------|-------------------------|--------------------------------------------------|
| NorESM2     | No                     | No                      | (Kirkevåg et al., 2018; Seland et al.,
2020)  |
| UKESM1      | Interactive            | Interactive             | (Archibald et al., 2019; Sellar et al.,
2019) |
| CNRM-ESM2-1 | No                     | Interactive             | (Michou et al., 2020)                            |
| MIROC6      | No                     | No                      | (Tatebe et al., 2019)                            |
| GFDL-ESM4   | Interactive            | Interactive             | (Horowitz et al., in prep)                       |
| CESM2-WACCM | Interactive            | Interactive             | (Gettelman et al., 2019)                         |
| GISS-E2-1   | Interactive            | Interactive             | (Bauer et al., 2020)                             |

Table 2 Sophistication of gas-phase chemistry used in the Earth system models (For further details see Thornhill et al. (submitted).

3.2 Model implementation of natural emissions of aerosols and ozone precursors.

**565 3.2.1 Land**

The principle land-based natural emissions analysed here are dust, and BVOCs and wetland methane (table 3).

Dust emissions are parameterised as a function of surface wind speeds or wind stress, and account for the amount of bare soil, soil type, and aridity (Ackerley et al., 2012; Collins et al., 2011; Evan et al., 2014; Fiedler et al., 2016; Huneeus et al., 2011; Shao et al., 2011; Zender et al., 2004). There is a variation between the models in the sizes considered, whether binned or

570 modal, and the optical properties of the dust particles (Kok et al., 2018; Xie et al., 2018). Table S1 lists the parameterizations for desert-dust aerosol for the contributing models and the simulated dust-aerosol sizes.

BVOC emissions are parametrised as a function of vegetation type and cover, and also temperature and photosynthesis rates (gross primary productivity) (Guenther, 1995; Pacifico et al., 2011; Sporre et al., 2019; Unger, 2014). Some parameterisations also include dependence on  $CO_2$  concentrations (Pacifico et al., 2012). Models differ in the speciation of the VOCs emitted

575 but typically include isoprene and monoterpenes, with different emission parameterisations for different species. The ability of VOCs to form secondary organic aerosol are typically parameterised as a fixed yield (Mulcahy et al., 2019). For further details see table S1 and references therein.

|                         | Dust                                                                                                                                          | B VOC                                                                                                                         | Wetland methane                                                                       |
|-------------------------|-----------------------------------------------------------------------------------------------------------------------------------------------|--------------------------------------------------------------------------------------------------------------------------------------|---------------------------------------------------------------------------------------|
| NorESM2                 | Interactive LAI, soil moisture, wind speed-varies                                                                                             | Dependence on PAR, temperature ,
LAI, vegetation type                                                                      | N/A                                                                            |
| UKESM1                  | Interactive vegetation (Interactive LAI, soil moisture, bare soil fraction)                                                     | Dependence on PAR, temperature,
vegetation                                                                                        | Dependent on
wetland fraction
available
substrate and
temperature:        |
| CNRM-
ESM2 -1 | Prescribed annual land cover (Séférian et al, 2019)                                                                             | Prescribed SOA
climatology <del>Prescribed</del>                                                                                  | N/A<del>Not</del>
ealculated
interactively                              |
| MIROC6                  | LAI from Land-surface model MATSIRO (Takata et al.
2003)-varies                                                                            | Prescribed                                                                                                                           | N/A                                                                            |
| GFDL-
ESM4           | Depends on simulated vegetation (LAI and SAI, used to calculate "bareness" fraction), land use, snow cover, wind speed Interactive vegetation | Externally prescribed LAI, vegetation
type and PAR <del>Dependence on PAR,
temperature.</del> Not dependent on

[revised manuscript text omitted]
-ESM2CNRM-ESM2-1, is consistent with the responses in 10mnear-surface-wind speed to *4xCO2* (figure S2S5). These clearly-reflect larger (smaller) increases in mean winds over regions where the mean emission amount is larger (smaller) for *4xCO2* compared to the pre-industrial climatology. The increase or decrease

in winds is also likely to be affected by changes in vegetation in semi-arid regions, e.g., the Sahel. As well as affecting the emissions, changing climate can also affect the removal of dust through changes in both dry and wet

deposition. In all models except UKESM1 the lifetime of dust increases (table 6). The effect of an increase in lifetime can be
 seen by comparing the change in AOD. The modelled changes in dust AOD in the *abrupt-4xCO2* experiment are one and a
 half to twice as large (for those models where lifetime increases) as would be expected assuming a linear scaling with emissions
 across all size ranges ("scaled AOD" in table 6).

The climate feedback parameter for dust ( $\alpha$ ) is given by the product of the radiative efficiencies ( $\phi$ ) with the sensitivities to climate ( $\gamma$ ). These vary from -0.016-012 to +0.048-0020 W m-2 K-1 with a multi-model mean of -0.0030026±0.008-0048 W m-2

- 645 2 K-1 i.e. averaging to a value nearconsistent with zero. Scaling with AOD change rather than emission change gives a slightly larger magnitude with a range -0.016 to +0.0048 W m-2 K-1 and a multi-model mean of -0.0040±0.0072 W m-2 K-1 Although some models obtain similar feedback terms, this is not necessarily for the same reason. For instance, CNRM ESM2 and UKESM1 have a positive dust feedback, though for opposite reasons; an increase in positive forcing in CNRM-ESM2 and a decrease in negative forcing in UKESM1. For instance GFDL-ESM4 and, NorESM2 have small feedback terms. NorESM2.
- 650 LM has a large ERF for doubled dust emissions, but athe small change in dust emission for 4xCO2, whereas GFDL-ESM4 has a large change in emissions but a small ERF however, does not lead to a large feedback for that model.
  Dust-aerosol feedback assessments are a relatively new area of research owing to the large uncertainties of climate models in simulating dust aerosols with changes in atmospheric composition. For instance, the spread in model estimates for dust aerosol changes in the 21st century is the largest among wildfires, biogenic SOA and DMS sulphate (Carslaw et al., 2010). Predictions
- 655 for future dust emission range from an increase (Woodward et al., 2005) to a decrease (Mahowald and Luo, 2003). The modelled feedbacks in table 6 are smaller in magnitude compared to the theoretical model estimates of -0.04 to +0.02 Wm-2 K-1 by Kok et al. (2018).